# A stochastic RNA editing process targets a select number of sites in individual *Drosophila* glutamatergic motoneurons

Andrés B Crane, Michiko O Inouye, Suresh K Jetti, J Troy Littleton*

The Picower Institute for Learning and Memory, Department of Brain and Cognitive Sciences, Department of Biology, Massachusetts Institute of Technology, Cambridge, United States

## eLife Assessment

This **important** study uses single-neuron Patch-seq RNA sequencing to investigate the process by which RNA editing can produce protein diversity and regulate function in various cellular contexts. The computational analyses of the data collected are **convincing**, and from an analytical standpoint, this paper is a notable advance in seeking to provide a biological context for massive amounts of data in the field. The study would be of interest to biologists looking at the effects of RNA editing in the diversification of cellular behaviour.

*For correspondence:
troy@mit.edu

**Abstract** RNA editing is a post-transcriptional source of protein diversity and occurs across the animal kingdom. Given the complete profile of mRNA targets and their editing rate in individual cells is unclear, single-cell RNA transcriptomes obtained by Patch-seq from *Drosophila* larval glutamatergic motoneuron subtypes were analyzed to determine the most highly edited targets and identify single neuron editing rules. 316 high-confidence A-to-I canonical RNA edit sites were identified, with 60 causing missense amino acid changes predicted to alter proteins regulating membrane excitability, synaptic transmission, or neuronal function. Twenty-seven canonical sites were edited at >90% frequency as observed for editing of mammalian AMPA receptors. However, most sites were edited at lower levels and generated variable expression of edited and unedited mRNAs, suggesting stochastic editing that may provide a mechanism to fine-tune synaptic function similar to alternative splicing. Noncanonical editing was also found to occur in these neurons, including a C-to-U edit that altered an amino acid in the capsid hinge domain of the synaptic plasticity regulator Arc1. Together, these data provide insights into how the RNA editing landscape may alter protein function to modulate the properties of two well-characterized neuronal populations in *Drosophila*.

## Introduction

RNAs undergo many post-transcriptional changes (*McCown et al., 2020*), with RNA editing being one of the more widely observed mRNA modifications. RNA editing alters single nucleotides in precursor mRNA and is most often catalyzed by the ADAR (adenosine deaminase acting on RNA) family of enzymes that bind double-stranded mRNA and irreversibly deaminate specific adenosines into inosines (A-to-I editing) (*Keegan et al., 2001*; *Nishikura, 2010*; *Rieder and Reenan, 2012*; *Shevchenko and Morris, 2018*). Inosines are recognized as guanines by the translation machinery, and edits within mRNA coding (CDS) exons can alter the protein's amino acid sequence. In addition, editing of 5' or 3' untranslated (UTR) exons can regulate mRNA splicing, translation, stability, or microRNA binding (*Berkovits and Mayr, 2015*; *Borchert et al., 2009*; *Lev Maor et al., 2007*; *Lopdell*

*et al., 2019*; *Mazloomian and Meyer, 2015*; *Nakano et al., 2017*; *Rueter et al., 1999*; *Shevchenko and Morris, 2018*; *Stellos et al., 2016*; *Yuan et al., 2023*). The levels of RNA editing (the fraction of total mRNA in an individual cell with a specific edit) depend on editing site, developmental stage, and cell type (*Birk et al., 2023*; *Buchumenski et al., 2017*; *Graveley et al., 2011*; *Kliuchnikova et al., 2020*; *Lundin et al., 2020*; *Ma et al., 2002*; *Picardi et al., 2017*; *Sapiro et al., 2019*; *Wahlstedt et al., 2009*). Although editing occurs across the animal kingdom, the biological significance of most editing events is unclear. Mammalian genomes encode two ADAR proteins (ADAR1 and ADAR2), with ADAR1 implicated in innate immunity (*Keegan et al., 2023*). The majority (>95%) of editing found in humans is mediated by ADAR1 and occurs within copies of the primate-specific non-coding *Alu* mobile retroelements that comprise ~11% of the human genome (*Athanasiadis et al., 2004*; *Bazak et al., 2014*; *Eisenberg and Levanon, 2018*; *Levanon et al., 2004*). ADAR2 has broader functional impacts through its ability to recode amino acids within a small number of proteins (*Keegan et al., 2023*; *Pinto et al., 2014*). RNA editing is particularly robust in neurons and several targets have been shown to regulate membrane excitability and synaptic function (*Alon et al., 2015*; *Brija et al., 2023*; *Brusa et al., 1995*; *Hoopengardner et al., 2003*; *Jepson and Reenan, 2009*; *Palladino et al., 2000a*; *Yablonovitch et al., 2017*).

In contrast to mammals, the *Drosophila* genome encodes a single ADAR that is functionally more similar to mammalian ADAR2 (*Palladino et al., 2000b*). Thousands of RNA editing sites have been identified in *Drosophila* using pooled RNA samples from whole animal or brain homogenates, enriched GFP-labeled cell populations obtained by flow cytometry, or through computational predictions of intron–exon base pairing (*Graveley et al., 2011*; *Hoopengardner et al., 2003*; *Keegan et al., 2023*; *Kliuchnikova et al., 2020*; *Mazloomian and Meyer, 2015*; *Ramaswami et al., 2013*; *Rodriguez et al., 2012*; *Sapiro et al., 2019*; *Stapleton et al., 2002*; *St Laurent et al., 2013*). Mutations in the sole *Adar* gene in *Drosophila* cause behavioral dysfunction, temperature-dependent seizures, and neurodegeneration in surviving adults (*Jepson and Reenan, 2009*; *Keegan et al., 2011*; *Khan et al., 2020*; *Li et al., 2014*; *Palladino et al., 2000a*; *Robinson et al., 2016*). Although ADAR is important in the *Drosophila* nervous system, the critical editing targets and functional impact of most edits remain unclear.

To examine the landscape of RNA editing more systematically, we determined the editing profile for *Drosophila* larval motoneurons (MNs) that are commonly used to study neuronal development and glutamatergic synapse biology (*Harris and Littleton, 2015*). Given the robust imaging and electrophysiology toolkits available to examine synaptic function at larval neuromuscular junctions, combined with GAL4 drivers for specific MN subtypes (*Aponte-Santiago et al., 2020*; *Pérez-Moreno and O'Kane, 2019*), these neurons provide an excellent model to define the functional impacts of RNA editing on neuronal communication. Two well-characterized glutamatergic MN subtypes innervate most larval body wall muscles. Tonic-like Ib neurons target single muscles and form weaker presynaptic active zones (AZs) that display synaptic facilitation, while phasic-like Is neurons synapse onto multiple muscles and form stronger AZs that undergo synaptic depression (*Genç and Davis, 2019*; *He et al., 2023*; *Jetti et al., 2023*; *Kurdyak et al., 1994*; *Lnenicka and Keshishian, 2000*; *Medeiros et al., 2024*; *Newman et al., 2017*; *Sauvola et al., 2021*). These MN subtypes also have diverse biophysical and morphological features (*Aponte-Santiago and Littleton, 2020*), providing an opportunity to identify the major targets for RNA editing and any cell-type-specific edits. We recently determined the transcriptomes for larval Ib and Is MN subtypes by performing single-cell Patch-seq RNA profiling from hundreds of individual Ib and Is neurons (*Jetti et al., 2023*). This allowed identification of differentially expressed genes (DEGs) that contribute to the unique properties of these neurons. In the current study, we extend this analysis to define the RNA editing landscape and single-cell editing rules for these two neuronal populations. These data identify the major targets for RNA editing in the two cell types and reveal a highly stochastic RNA editing process that occurs across individual MNs from both populations.

## Results

### Single MN RNA editing suggests functional diversification of a select number of target neuronal proteins

Canonical RNA editing occurs when ADAR deaminates adenosines, converting them into inosines that are interpreted as guanosine by the translation machinery (*Figure 1A*). To determine specific sites that are edited in *Drosophila* third instar larval MN subpopulations, we analyzed single-cell RNA sequencing data obtained by Patch-seq profiling from 105 individual MN1-Ib (hereafter referred to as Ib) and 101 individual MNISN-Is (hereafter referred to as Is) neurons labeled with GFP using GAL4 drivers specific to each cell type (*Jetti et al., 2023*). Candidate RNA editing sites were obtained using GATK (*McKenna et al., 2010*; *Poplin et al., 2017*) and SAMtools (*Danecek et al., 2021*) to compare genomic DNA sequenced from parental strains to their single-cell RNA transcriptomes to identify base pair mismatches that did not represent genomic single-nucleotide polymorphisms (SNPs; *Figure 1B, C*). To focus on high-confidence RNA editing events and further exclude potential sequencing errors that could appear as very low abundance edits, DNA–RNA mismatches were filtered and excluded if the edited site was observed at low levels (<10% editing) or present in only a few neurons (<10 Ib or Is cells). After filtering, 316 high-confidence canonical A-to-I editing sites were identified across both MN transcriptomes (*Figure 1D*, *Supplementary files 1 and 2*). A number of C-to-T and G-to-A transversions were also observed (*Figure 1D*, *Supplementary files 3 and 4*), suggesting noncanonical editing occurs in MNs as seen in other *Drosophila* cell types (*Sapiro et al., 2019*; *Stapleton et al., 2006*). Although noncanonical editing has been described (*Bhakta and Tsukahara, 2022*; *Gott and Emeson, 2000*; *McCown et al., 2020*; *Pecori et al., 2022*), the biological significance and molecular pathways mediating these edits are less clear. We largely focused on canonical A-to-I editing, given it is likely to have a greater impact on MN function, in contrast to noncanonical editing that appears less impactful based on the target sites identified (see analysis of noncanonical edits).

The 316 canonical edits were distributed across 210 genes and included previously unannotated sites based on Flybase (*Supplementary file 2*). 55% (175/316) of canonical edits occurred in mRNA coding sequences (CDS), while 36% (115/316) were in 3′UTR and 8% (26/316) in 5′UTR (*Figure 1E*), consistent with prior editing distributions in *Drosophila* (*St Laurent et al., 2013*). The editing rate within each region of the mRNA was normally distributed (p < 0.05, Kolmogorov–Smirnov test). Mean editing rate in the CDS (0.71) was significantly higher (ANOVA $F$ statistic = 20.171, degrees of freedom (df) = 2, Tukey $p < 7.8 \times 10^{-10}$) than 3′UTR (0.59) and was also higher than 5′UTR (0.64) editing (*Figure 1F*). For CDS edits, 58% (102/175) resulted in a missense amino acid substitution. Given that edits that result in significant amino acid changes are more likely to alter protein function, we analyzed these edits in more detail. Missense changes were classified as significant if they altered the residue classification between (but not within) positive (R, K) or negative (D, E), hydrophobic (A, V, I, L, M, F, Y, W), polar (N, Q), potential phosphorylation sites (S, T), or structural residues (C, G, P, H). Based on these criteria, 59% (60/102) of missense edits resulted in significant alterations to the affected amino acid. The most common missense change was L>M, while the most common disruptive alteration was change of an existing residue into a glycine (S>G, E>G, *R*>G, and D>G, *Figure 1G*). No canonical edit was identified that induced a premature stop codon. However, an edit in *Cnep1r2*, which encodes a serine/threonine phosphatase involved in nuclear pore formation and lipid metabolism (*Jacquemyn et al., 2021*), changed a stop codon into a tryptophan residue. The unedited transcript encodes a protein that is conserved with its mammalian homologs through the C-terminus, while the edited transcript generates a modified protein with 14 additional C-terminal amino acids.

### Editing rules from single neuron analysis

Given the RNA editing data derived from single neurons with potentially distinct RNA editing patterns, we were interested in defining general rules for single-cell editing compared to those observed using bulk RNAseq approaches. Early studies of editing rates for mammalian GluR2 AMPA receptors observed either a lack of editing or complete editing of *GluR2* mRNA depending on developmental stage, suggesting an all-or-none editing model. However, recent analyses indicate editing rates for most sites are more variable (*Graveley et al., 2011*; *Rodriguez et al., 2012*; *Sapiro et al., 2019*; *St Laurent et al., 2013*). To examine the editing profile across every editing site in individual Ib and Is neurons, the average fraction of edited reads in single cells was determined (*Figure 1F*). Editing rates spanned from the 10% threshold to 100% across both neuronal populations. The average editing

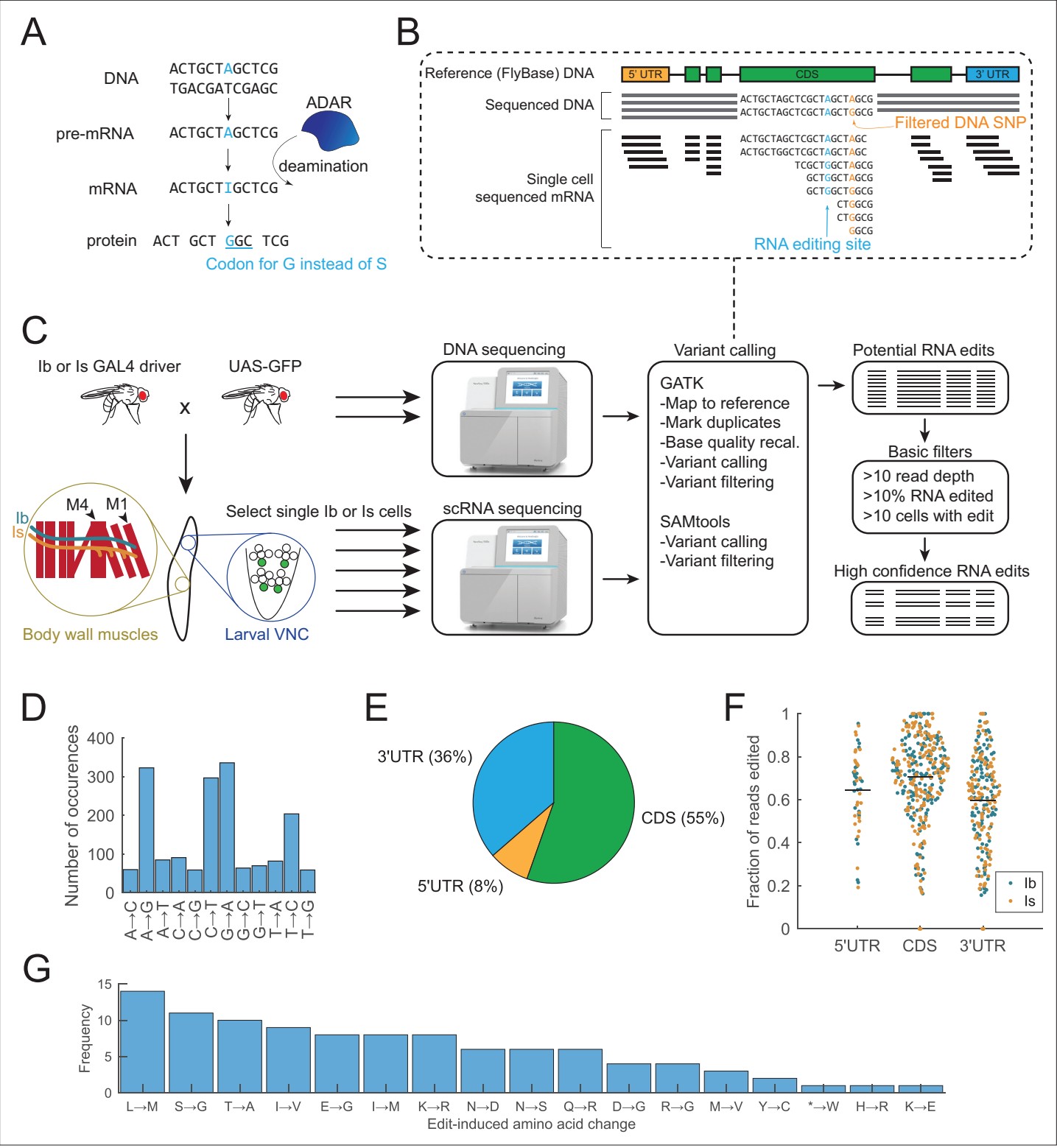

**Figure 1.** RNA editing in larval motoneurons (MNs) occurs primarily in mRNA coding regions and displays variable editing rates per target site. (**A**) Schematic of canonical RNA editing where ADAR deaminates an adenosine, forming an inosine, which is interpreted as guanosine by translation machinery. RNA editing in exons can cause a missense mutation. (**B**) RNA edits (blue nucleotides) were identified by comparing sequenced DNA and RNA for mismatches. Single-nucleotide polymorphisms (SNPs, orange nucleotides) were filtered from the data. Gray and black horizontal bars represent DNA and RNA sequencing reads aligned to the reference genome. (**C**) Ib- or Is-specific Gal4 drivers were used to express UAS-GFP to label the cells in the third instar ventral nerve cord. Cells were individually patched and their RNA extracted for single-cell RNA sequencing. Genomic DNA from the

*Figure 1 continued on next page*

*Figure 1 continued*

parental Gal4 and UAS lines was DNA sequenced. Variants were called using the Genome Analysis Toolkit (GATK) and SAMTools, with the intersection of the variants identified as potential RNA edits. Potential RNA edits were further filtered by read depth, editing percentage, and number of cells using a custom MATLAB code to result in a final list of high-confidence edit sites. (**D**) Among the 1637 high-confidence edit sites, 316 were canonical A-to-G edits. C-to-T and G-to-A editing was also abundant. (**E**) The majority of editing sites are localized in the mRNA CDS region. (**F**) Edits found in the CDS had a higher average editing rate than those in the 5′UTR or 3′UTR. Each dot represents mean editing of Ib or Is cells at a single genomic location. (**G**) The frequency distribution of missense mutations from canonical editing in Ib and Is MNs is shown.

rate per site was 66% and was not significantly different between Ib and Is neurons (*Figure 1F*, p = 0.94, Student's *t*-test, *t* statistic = 0.8, df = 630). Consistent with prior studies in adult *Drosophila* CNS (*Hoopengardner et al., 2003*; *Palladino et al., 2000a*; *Sapiro et al., 2019*), mRNAs encoding regulators of neuronal excitability and function were over-represented as editing targets in larval MN transcriptomes (*Figure 2A*). The largest number of edits (15 sites) within a single gene occurred in the mRNA encoding Complexin (Cpx), a key regulator of synaptic vesicle (SV) fusion (*Buhl et al., 2013*; *Cho et al., 2015*; *Cho et al., 2014*; *Huntwork and Littleton, 2007*; *Jorquera et al., 2012*). Other highly edited mRNAs included those encoding voltage-gated ion channels (Cac, Para, Shab, Sh, Slo), ligand-gated ion channels (nAChRalpha5, nAChRalpha6, nAChRalpha7, nAChRbeta1, Rdl), and regulators of synaptic transmission (EndoA, Syx1A, Rim, RBP, Vap33, Lap).

RNA editing events that induce functional changes to a transcript would be predicted to be expressed in most cells. Although no edit site was detected in 100% of sampled neurons, often due to low read coverage at specific edit sites in any given cell, some edits were found in as many as 87% (175/206) of neurons. On average, an edited site was detected in ~19% (~40/206) of neurons (*Figure 2B*). Edit sites observed in the largest fraction of sampled neurons are shown in *Figure 2C*. Edits in the 5′ and 3′UTR were more common in these cases, with edits in *ATPsynCF6* and *Syx1a* found in >80% of individual neurons. There was also a tendency for edit events to occur in proximity to other edits. Out of 316 sites, 53% had a nearby edit within 100 nucleotides and 26% had a nearby edit with 10 nucleotides (*Figure 2D*), suggesting favorable ADAR binding to some dsRNA regions increases editing rates at nearby adenosines (*Buhl et al., 2013*). Indeed, 12 edits with an average editing rate of 45% occurred in a stretch of 118 nucleotides in the 3′UTR of *Cpx* (3R:4301140–4301258). In summary, these data indicate RNA editing in larval MNs acts on a small number of target mRNAs and ranges from low rates of editing to sites where the mRNA is fully edited.

## Identification of editing sites that alter conserved amino acids

Amino acid changes that occur in conserved regions of a protein are more likely to have significant impact on function. We filtered significant missense changes caused by editing and focused on those occurring in conserved residues or protein domains based on homology across other invertebrate or vertebrate species. 43 editing-induced significant missense changes were identified in conserved areas across several protein classes, including those regulating membrane excitability, synaptic transmission, and cell signaling (*Table 1*). 49% (21/43) of the edited residues altered amino acids that were conserved in vertebrates and invertebrates, while the remaining 51% (22/43) changed residues conserved across other insect homologs. Notably, many of the edits occurred within annotated protein domains. Given these edits, which alter conserved amino acids and are likely to have a larger impact on larval MN function, they were examined in more detail.

Multiple subunits of the nicotinic acetylcholine receptor (nAChR) family, which function as ligand-gated cation channels mediating fast excitatory neurotransmission, were found to undergo editing. One edit in nAChRalpha5 converts a threonine to alanine (T791A) in the fourth transmembrane domain and is edited at very high levels (Ib: 99%, Is: 96%, *Figure 3A*), suggesting the edited version represents the majority of nAChRalpha5 protein in MNs. Edits in nAChRalpha6 (*Figure 3B*) and nAChRbeta1 (*Figure 3C*) altered conserved residues in the extracellular ligand-binding domains of the proteins. Editing also occurred in several proteins regulating neuronal excitability, including E-to-G and Y-to-C edits in the ATPα sodium pump (*Figure 3D*), a highly edited S-to-G site (Ib: 86%, Is: 91%) in the ninth transmembrane segment of the voltage-gated calcium channel Cacophony (*Figure 3E*), and multiple edits in the ion transport domain of other voltage-gated ion channels, including Shab, Sh, Slo, and Para (*Figure 3F*).

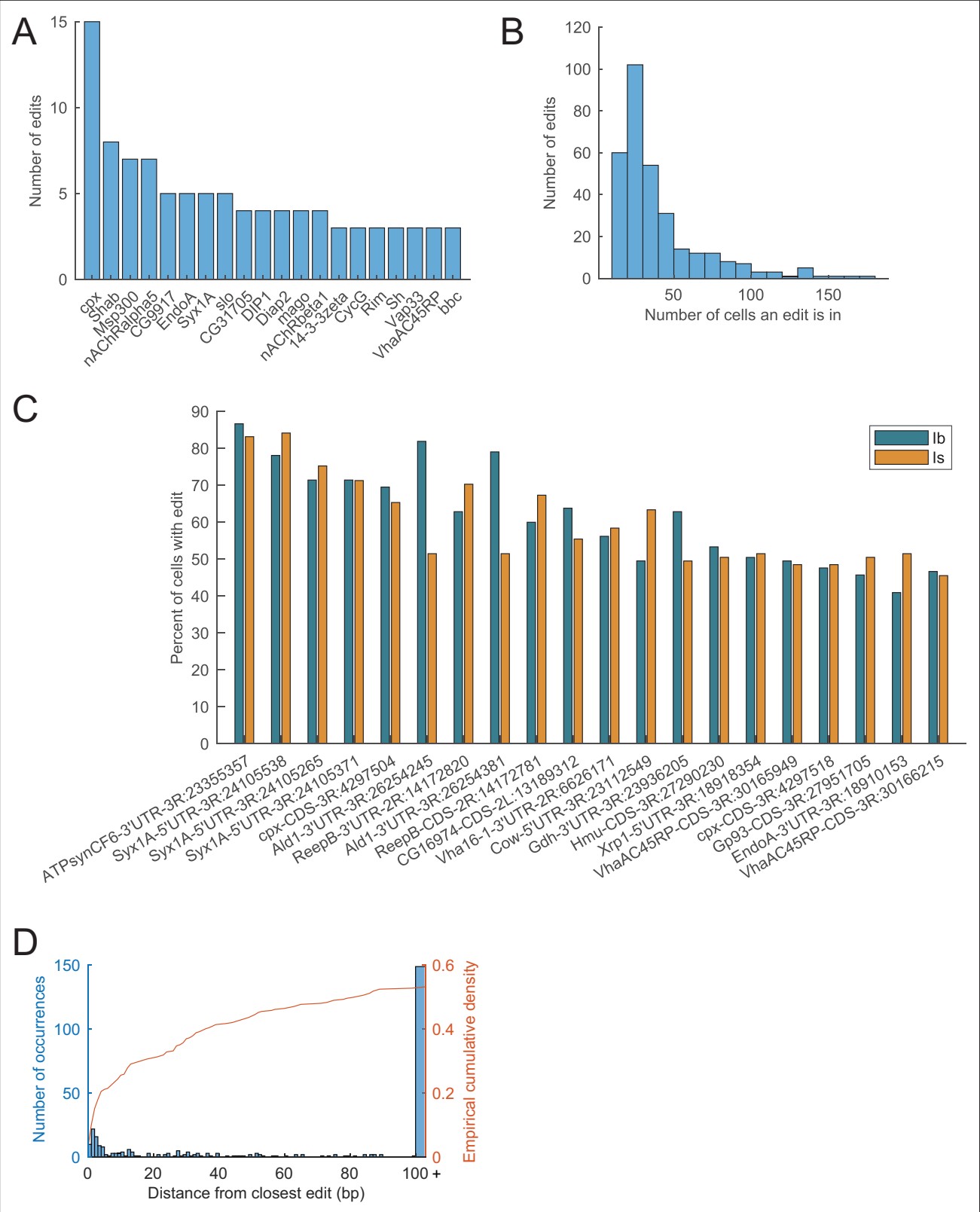

**Figure 2.** The most highly edited targets include mRNA encoding neuronal and synaptic proteins. (**A**) The 20 genes with the most editing sites are enriched for those regulating excitability and neurotransmission. (**B**) Distribution of the frequency of edits across the neuronal population sampled. (**C**) The top 20 edits that were most frequently identified in cells are shown. (**D**) Distribution of edits based on proximity to other editing sites (blue histogram) with the cumulative probability density denoted (orange line). Events more than 100 nucleotides away are shown combined in a single histogram bar.

**Table 1.** Canonical RNA edits altering conserved amino acids in target proteins.

Canonical RNA edit sites that caused significant missense mutations that alter a conserved amino acid across other insect or mammalian homologs are listed. Description of column headers: Gene – name of gene; Flybase ID – Flybase ID of the gene; Edit position – position of the edit in the genome, formatted as chromosome:position; Previously known RNA editing site – based on publicly available annotations in FlyBase, whether this genomic position has been previously annotated as having RNA editing activity; Pfam domain – based on FlyBase annotations, whether this genomic position falls into an annotated protein domain; Edit site conservation – insects indicate the unedited amino acid created by this codon is conserved in insects; mammals indicate the unedited amino acid created by this codon is conserved in mammals; Original codon – the unedited codon; Edited codon – codon after editing; Original AA – the amino acid produced from the unedited RNA; Edited AA – the amino acid produced from the edited RNA; Gene function – brief description of the gene's predicted function.

| Gene | Flybase ID | Edit position | Previously known RNA editing Site | Pfam domain | Edit site conservation | Original codon | Edited codon | Original AA | Edited AA | Gene function |
|---|---|---|---|---|---|---|---|---|---|---|
| **Membrane excitability** | | | | | | | | | | |
| qvr | FBgn0260499 | 2R:11447607 | Yes | QVR | Insects | AGT | GGT | S | G | Regulates K$^+$ channels |
| Shab | FBgn0262593 | 3L:2925255 | Yes | T1-type_BTB | Insects | AGT | GGT | S | G | K$^+$ channel |
| Shab | FBgn0262593 | 3L:2941312 | Yes | Ion_trans_dom | Mammals | ACA | GCA | T | A | K$^+$ channel |
| Shab | FBgn0262593 | 3L:2941364 | Yes | Ion_trans_dom | Mammals | TAT | TGT | Y | C | K$^+$ channel |
| Shab | FBgn0262593 | 3L:2941396 | Yes | Ion_trans_dom | Insects | ACT | GCT | T | A | K$^+$ channel |
| CG1090 | FBgn0037238 | 3R:4365385 | Yes | NaCa_Exmemb | Mammals | AGC | GGC | S | G | Na$^+$/Ca$^{2+}$ (NCX) exchange |
| Atpalpha | FBgn0002921 | 3R:20964328 | Yes | - | Mammals | GAG | GGG | E | G | Na$^+$ pump |
| Atpalpha | FBgn0002921 | 3R:20965039 | Yes | - | Insects | TAC | TGC | Y | C | Na$^+$ pump |
| slo | FBgn0003429 | 3R:24675663 | Yes | Ion_trans_dom | Mammals | AAT | GAT | N | D | K$^+$ channel |
| OtopLa | FBgn0259994 | X:5631048 | Yes | Otopetrin | Insects | ACT | GCT | T | A | Proton (H$^+$) channel |
| para | FBgn0285944 | X:16464719 | Yes | Ion_trans_dom | Insects | AAC | AGC | N | S | Na$^+$ channel |
| para | FBgn0285944 | X:16482682 | Yes | Ion_trans_dom | Insects | AAT | GAT | N | D | Na$^+$ channel |
| Sh | FBgn0003380 | X:17930658 | Yes | - | Mammals | ACG | GCG | T | A | K$^+$ channel |
| **Presynaptic function** | | | | | | | | | | |
| cpx | FBgn0041605 | 3R:4297517 | Yes | Synaphin | Mammals | AAT | GAT | N | D | SV exocytosis |
| cpx | FBgn0041605 | 3R:4297518 | Yes | Synaphin | Mammals | AAT | AGT | N | S | SV exocytosis |
| lap | FBgn0086372 | 3R:7192775 | Yes | - | Insects | ACA | GCA | T | A | SV endocytosis |
| Rbp | FBgn0262483 | 3R:15399446 | Yes | - | Insects | ACC | GCC | T | A | AZ protein |
| Rim | FBgn0053547 | 3R:17881419 | No | - | Insects | AGA | GGA | R | G | AZ protein |
| EndoA | FBgn0038659 | 3R:18907675 | Yes | BAR_dom | Mammals | AAG | GAG | K | E | SV endocytosis |
| VhaAC45RP | FBgn0051030 | 3R:30165949 | No | - | Insects | GAG | GGG | E | G | Part of SV proton pump |

*Table 1 continued on next page*

*Table 1 continued*

| Gene | Flybase ID | Edit position | Previously known RNA editing Site | Pfam domain | Edit site conservation | Original codon | Edited codon | Original AA | Edited AA | Gene function |
|---|---|---|---|---|---|---|---|---|---|---|
| cac | FBgn0263111 | X:11971853 | Yes | Ion_trans_dom | Insects | AGT | GGT | S | G | Ca$^{2+}$ channel |
| **Postsynaptic function** | | | | | | | | | | |
| nAChRalpha6 | FBgn0032151 | 2L:9809349 | Yes | Neur_chan_lig-bd | Insects | AAC | AGC | N | S | nAch receptor |
| nAChRalpha6 | FBgn0032151 | 2L:9809350 | Yes | Neur_chan_lig-bd | Insects | AAC | GAC | N | D | nAch receptor |
| nAChRalpha5 | FBgn0028875 | 2L:14089204 | Yes | Neurotrans-gated_channel_TM | Mammals | ACA | GCA | T | A | nAch receptor |
| Prosap | FBgn0040752 | 2R:14062651 | Yes | - | Insects | CAG | CGG | Q | R | Shank – PSD protein |
| nAChRbeta1 | FBgn0000038 | 3L:4432487 | Yes | Neur_chan_lig-bd | Insects | AGA | GGA | R | G | nAch receptor |
| **Cell metabolism** | | | | | | | | | | |
| CG1265 | FBgn0035517 | 3L:4255132 | No | - | Mammals | GAG | GGG | E | G | Lysosomal Solute Transporter |
| **Cell signaling** | | | | | | | | | | |
| bsk | FBgn0000229 | 2L:10248292 | No | Prot_kinase_dom | Mammals | GAC | GGC | D | G | JNK protein kinase |
| CalpB | FBgn0025866 | 3L:9892134 | Yes | - | Mammals | AAT | AGT | N | S | Ca$^{2+}$-activated protease |
| **Cytoskeletal or cell–cell adhesion** | | | | | | | | | | |
| betaTub56D | FBgn0284243 | 2R:19447593 | No | - | Mammals | GAC | GGC | D | G | Tubulin component |
| CG13506 | FBgn0034723 | 2R:22391510 | Yes | - | Insects | CAG | CGG | Q | R | Ig domain protein |
| CG32264 | FBgn0052264 | 3L:3717400 | No | - | Mammals | AGG | GGG | R | G | Role in actin biology |
| **Membrane trafficking** | | | | | | | | | | |
| Snx21 | FBgn0031457 | 2L:2764358 | No | - | | GAG | GGG | E | G | Sorting Nexin |
| Sec23 | FBgn0262125 | 3R:5650231 | Yes | Sec23/24_trunk_dom | Mammals | CAG | CGG | Q | R | COPII component in ER–Golgi transport |
| Sec23 | FBgn0262125 | 3R:5650351 | Yes | Sec23/24_trunk_dom | Mammals | CAG | CGG | Q | R | COPII component in ER–Golgi transport |
| CG42613 | FBgn0261262 | 3R:18969789 | No | CUB_dom | Insects | GAT | GGT | D | G | Plasma membrane protein |
| **Mitochondrial function** | | | | | | | | | | |
| muc | FBgn0283658 | 2L:7429847 | No | 2-oxoacid_DH_acylTfrase | Mammals | GAG | GGG | E | G | Component of mitochondrial pyruvate dehydrogenase complex |
| Opa1 | FBgn0261276 | 2R:14234798 | No | - | Mammals | AAT | GAT | N | D | Mitochondrial dynamin |

*Table 1 continued on next page*

*Table 1 continued*

| Gene | Flybase ID | Edit position | Previously known RNA editing Site | Pfam domain | Edit site conservation | Original codon | Edited codon | Original AA | Edited AA | Gene function |
|---|---|---|---|---|---|---|---|---|---|---|
| bonsai | FBgn0026261 | 2R:22638356 | No | Ribosomal_uS15 | Insects | GAC | GGC | D | G | Mitochondrial ribosomal protein |
| **Transcription/splicing/nuclear function** | | | | | | | | | | |
| RNaseZ | FBgn0028426 | 2R:10307375 | No | - | Mammals | GAA | GGA | E | G | tRNA processing |
| Hr78 | FBgn0015239 | 3L:21494397 | No | Nucl_hrmn_rcpt_lig-bd | Mammals | GAG | GGG | E | G | Nuclear steroid receptor |
| beag | FBgn0037660 | 3R:9063537 | No | RED_C | Insects | AGC | GGC | S | G | Splicesomal component – regulates Fas2 splicing |
| Sxl | FBgn0264270 | X:7085816 | Yes | - | Insects | AGC | GGC | S | G | RNA-binding protein |

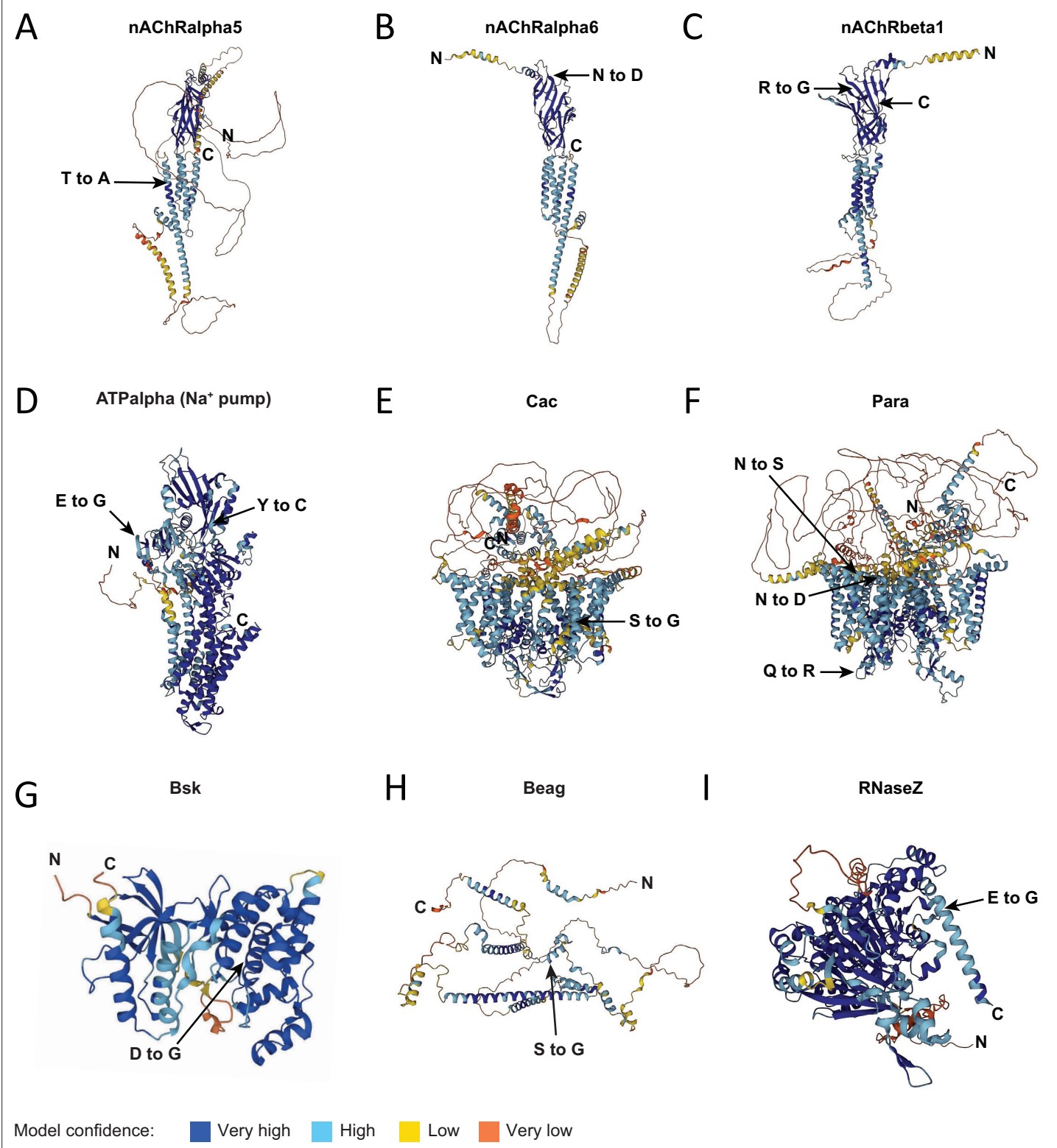

**Figure 3.** Location of selected missense RNA edits altering conserved residues within ion channels and signaling proteins. AlphaFold predicted structures with RNA edit locations marked for selected postsynaptic receptors (**A–C**), excitability proteins (**D–F**), and Bsk (**G**), Beag (**H**), and RNaseZ (**I**). Color indicates model confidence at individual residues: dark blue (very high), light blue (high), yellow (low), and orange (very low). The N-terminus (N) and C-terminus (C) are denoted for each protein.

Beyond neuronal excitability, mRNAs encoding proteins regulating the SV cycle were also editing targets (*Table 1*). Three edit sites within the C-terminal domain of the Cpx7A splice isoform were identified and previously described (*Brija et al., 2023*). Editing of adjacent adenosines encoding the N130 amino acid produces N130S, N130G, or N130D variants that alter Cpx's ability to regulate SV fusion (*Brija et al., 2023*). Editing was also observed in a key regulator of endocytosis, Endophilin A (EndoA), resulting in a change from a positively charged K to a negatively charged E residue at amino acid 140 within the F-BAR lipid-binding domain. Given the K residue is conserved in all Endophilin homologs, one would predict this charge substitution alters EndoA function in endocytosis within MNs. Editing of another SV endocytosis regulator, the Clathrin AP180 adaptor homolog Lap, occurred at high rates (Ib: 88%, Is: 95%) and resulted in a T-to-A change in a conserved residue. Edits were also identified in the AZ proteins Rim and RBP that altered C-terminal residues conserved in other insect homologs. Similarly, an edit in the VhaAC45RP subunit of the SV proton pump resulted in neutralization of a negatively charged E residue conserved in other insect homologs.

Within the group of edits predicted to significantly impact protein function, 67% (29/43) were previously annotated on FlyBase as editing sites, while 33% (14/43) were not previously described. Newly identified sites could be enriched targets for third instar larval MNs or missed in prior work. Among the new edits was a missense D-to-G change in Bsk (*Figure 3G*), a conserved serine/threonine protein kinase central to the stress-induced c-Jun N-terminal kinase (JNK) pathway. The edited D residue is universally conserved in JNK kinase homologs and resides within the core of the kinase domain. Editing from D to G occurs at moderate levels (Ib: 34%, Is: 30%) and is predicted to be disruptive to the protein's function. Another novel editing site was identified in the spliceosomal protein Beag (*Figure 3H*). Beag forms a complex with Smu1 and additional alternative splicing proteins and is required for larval synapse development and function (*Beck et al., 2012*). The edited serine residue is conserved across insect species and is edited at high rates within MNs (Ib: 72%, Is: 57%). The S-to-G substitution occurs in the RED-like protein C-terminal domain, a region involved in nuclear localization and binding to other nuclear-localized proteins (*Ulrich et al., 2016*). A third newly identified edit site results in an E-to-G substitution in the highly conserved endoribonuclease RNase Z (*Figure 3I*) involved in tRNA processing (*Saoura et al., 2017*). The E744 residue is universally conserved in invertebrate and vertebrate homologs and localizes downstream of its metallo-beta-lactamase C-terminal domain. These data indicate RNA editing alters highly conserved residues across several proteins, including those involved in membrane excitability and synaptic transmission that are likely to impact the development or function of larval MNs.

## Characterization of highly edited transcripts and editing stochasticity

We next examined sites with a very high editing rate that also induced missense amino acid substitutions to identify edits that may cause larger functional impacts on MNs similar to mammalian AMPA receptor editing. Twenty-seven sites in eighteen genes were found to be edited at >90% levels in Ib or Is neurons that underwent editing at the location (*Supplementary file 5*). 85% (23/27) of these sites were previously annotated on FlyBase, consistent with these targets being more commonly identified given their high editing rate. Highly edited sites were overrepresented in genes regulating membrane excitability and synaptic function. Ten sites in nine genes displayed 100% editing in at least one of the two neuronal populations, including (1) *quiver*, a trafficking regulator of K$^+$ and nACh receptors; (2, 3) *nAChRalpha5* and *nAChRalpha6*, subunits of the nicotinic acetylcholine receptor; (4) *Prosap*, the *Drosophila* SHANK synaptic scaffolding homolog; (5) *Rdl*, a GABA receptor; (6) *Nckx30C*, a Na$^+$/K$^+$-dependent Ca$^{2+}$ exchanger; (7) *Shab*, a delayed rectifier K$^+$ channel; (8) *Ythdc1*, a regulator of alternative splicing; and (9) *RNaseZ*, a pre-tRNA endoribonuclease. Although these sites were highly edited, most sites showed more variable patterns of editing, with some neurons editing the site at high levels while others had lower rates or no editing. *Adar* mRNA levels were not significantly different between Ib and Is neurons (*Figure 4A*, p = 0.55, adjusted p-value from RNA sequencing Wald statistic; *Jetti et al., 2023*). Similar to prior studies (*Graveley et al., 2011*; *Lundin et al., 2020*; *Picardi et al., 2017*; *Wahlstedt et al., 2009*), *Adar* mRNA expression level at the resolution of single cells did not correlate with overall editing rates (black line in *Figure 4B*). Indeed, neurons with low or high *Adar* transcript levels were able to edit specific sites up to 100%. However, we noticed that edit sites in abundantly expressed mRNAs (expression level in the top 30% as noted by red dots in *Figure 4B*) generally had lower editing rates. Given that 4000-fold differences in gene expression can occur

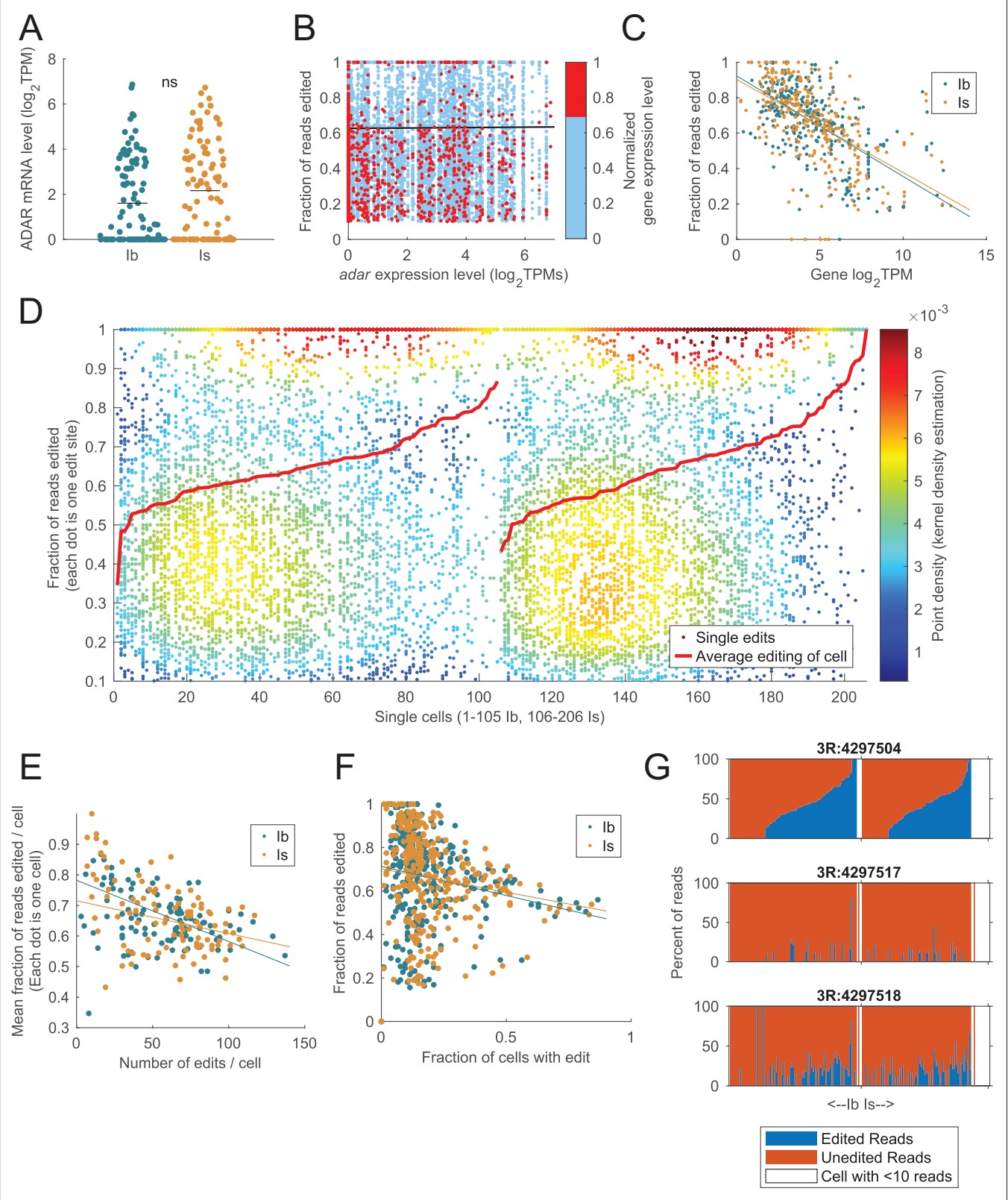

**Figure 4.** Adar activity becomes rate-limiting for abundant mRNA transcripts. (**A**) *Adar* transcript expression levels for individual Ib and Is neurons are not significantly different. (**B**) *Adar* expression did not correlate with editing level. Genes in the top 30% of transcript expression levels are shown in red. (**C**) RNA editing level at single sites is inversely correlated with the mRNA expression level of the target gene for both Ib and Is neurons. (**D**) Distribution of the editing rate for each individual target site (single dots) arranged in columns corresponding to the individual neuron. Neurons are ordered from

*Figure 4 continued on next page*

*Figure 4 continued*

lowest to highest average editing level per cell within the Ib and Is subtypes. Color denotes the density of dots in the graph, where warm colors indicate high density and cool colors low density. Single cells did not edit all sites in their transcriptome equally. Rather, target sites within a single cell were edited at levels spanning from the 10% threshold to 100% edited. (**E**) There was an inverse correlation between the number of edits in a given cell and mean editing level, consistent with a model where ADAR becomes rate-limiting for abundant mRNAs. Each dot represents a single cell. (**F**) Editing level is inversely correlated with the fraction of cells containing the edit. (**G**) Variability in editing levels for three Cpx sites highlighting cells with completely unedited (red columns), partially edited (blue and red columns), or fully edited (blue columns) transcript. Each column represents a single cell, and column order is held constant in each subplot to highlight how editing in neighboring genomic locations can vary within a single cell. White columns represent cells lacking enough reads at the site for analysis.

across transcripts within larval MNs (*Jetti et al., 2023*), we hypothesized that abundant transcripts might result in lower editing if ADAR activity becomes rate-limiting. Indeed, mRNA expression level was inversely correlated to the fraction of reads edited at individual sites for both Ib and Is subtypes (Ib: Pearson's $r = -0.59$, $r^2 = 0.35$; Is: $r = -0.58$, $r^2 = 0.34$, *Figure 4C*). We conclude that editing levels are not directly limited by ADAR expression but rather by target gene expression, suggesting ADAR can become rate-limiting when high levels of an editable pre-mRNA transcript are present.

Although *Adar* mRNA levels did not correlate with editing rate, it is possible the expression of other RNA-binding proteins or editing factors impacts the rate of editing. If so, individual neurons that express higher levels of these factors would be expected to display higher overall editing rates. To examine this possibility, editing rate for every target site was plotted for each individual Ib and Is MN (*Figure 4D*). Neurons from both populations displayed a range of average editing rates across all target sites that varied from ~30% to >80%, indicating the presence of 'low' and 'high' editors (*Figure 4D*). Although it is unclear what drives these differences in editing rate for neurons of the same cell type, future analysis of DEGs between these groups might identify candidate proteins that impact editing across an otherwise homogenous neuronal population. Another possibility for differential editing is that cells with the highest editing rate edit fewer transcripts overall but at a higher rate per transcript. Consistent with this model, the mean fraction of reads edited versus the number of edits across individual neurons displayed a negative correlation (Ib: Pearson's $r = -0.33$, $r^2 = 0.11$; Is: $r = -0.50$, $r^2 = 0.25$, *Figure 4E*). We hypothesized that an edit found in most cells would potentially have a high editing rate. However, a negative correlation was observed, suggesting the more cells an edit was found in, the lower the mean editing rate (*Figure 4F*). Beyond variability in overall editing rate, some individual edit sites displayed a highly stochastic cell-to-cell editing rate whereby some cells expressed a mixture of edited and unedited mRNA, whereas others had all or none editing for that same target site. Several representative regions showing stochastic editing rates are shown for three such edits in *Cpx* (*Figure 4G*). In summary, editing rates for most target sites are variable at the level of single neurons and individual edited adenosine residues, with only a small number of sites undergoing complete editing.

## Comparison of RNA editing between Ib and Is MN populations

Given structural and functional differences between tonic Ib and phasic Is MNs, we hypothesized unique RNA editing targets may help establish their distinct features. However, the overall landscape of RNA editing across Ib and Is neurons was similar at the level of edit site location (5′UTR, CDS, 3′UTR) and editing rate (*Figure 1F*). In addition, the number of edit sites and fraction of reads edited per cell showed similar profiles (*Figure 5A*). A comparison of editing rate at individual sites indicated most were edited to the same level between Ib and Is neurons (Pearson's $r = 0.73$, $r^2 = 0.53$, *Figure 5B*). The fraction of Ib and Is MNs that expressed individual edits was also correlated (Pearson's $r = 0.83$, $r^2 = 0.69$), although several edits were only detected in one of the two cell types (red and blue dots in *Figure 5C*). Though editing rules were generally similar between the two populations, several differences emerged. The lower tail of the distribution for the fraction of edited reads displayed a sharper drop across Ib cells (*Figure 4D*), indicating a subset of these neurons has distinctly lower editing rates than observed in Is neurons. Similarly, the upper tail of the distribution revealed a subset of Is neurons with higher editing rates than observed within the Ib population (*Figure 4D*).

To identify sites with significantly different editing rates between subtypes, differentially expressed mRNAs were removed from the analysis. Following DEG filtering, 26 RNA edit sites across 22 genes were identified that displayed statistically significant editing differences (*Figure 5D, E*, *Supplementary*

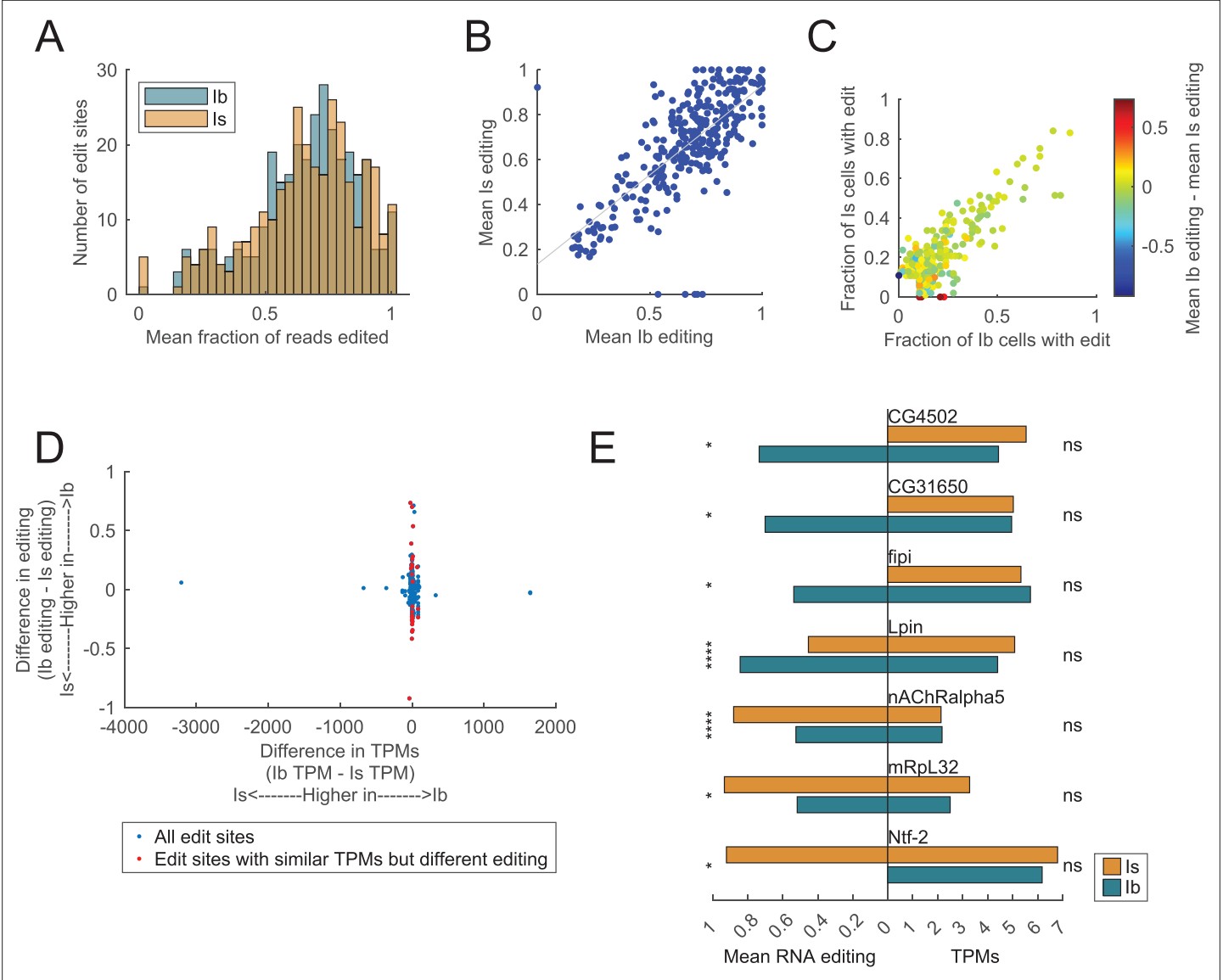

**Figure 5.** Editing targets and editing rate are similar across Ib and Is motoneuron subtypes. (**A**) Average RNA editing level between Ib and Is neurons was not significantly different. (**B**) Levels of Ib and Is RNA editing at individual sites were highly correlated. Dots along the axes are from sites with editing in one cell type but not the other. (**C**) The fraction of Ib or Is cells expressing a given edit was highly correlated. Color indicates the difference in editing rate for each edit, with warm colors representing a higher editing rate in Ib and cooler colors a higher editing rate in Is. (**D**) Edit sites are graphed according to differences in gene expression and RNA editing level between Ib and Is. A few edits were in differentially expressed genes (DEGs; dots spread horizontally), but most genes had relatively little mRNA expression differences (dots clustered near 0 transcripts per million (TPMs)). Genes with significant differences between Ib and Is RNA editing levels (dots spread vertically), but non-significant gene-level expression differences are highlighted in red. (**E**) Twenty-six RNA editing sites (red dots in (D)) displayed statistically significant RNA editing differences between Ib and Is that did not represent DEGs. The mean RNA editing level (left side) is contrasted with gene expression level (right side) for several representative edit sites. Statistical significance for RNA editing and gene expression level is shown to the left and right of each edit site, respectively.

file 6). Although 3′UTR edits comprised only 36% of all editing sites (*Figure 1B*), 46% (12/26) of differentially edited sites between Ib and Is neurons were in 3′UTRs. Given 3′UTR editing can modify microRNA-binding sites or protein translation rates, it is interesting to consider if post-transcriptional regulation through RNA editing in this region differentially impacts protein production. Of particular interest are two 3′UTR sites in *complexin* (3R:4301150 and 3R:4301158) that are edited at greater rates in Is neurons, which are known to express lower Cpx protein levels at synapses (*Newman et al., 2022*). We conclude that differential editing between Ib and Is neurons that alters amino acid identity is rare and unlikely to play a major role in the morphological and functional diversity between these

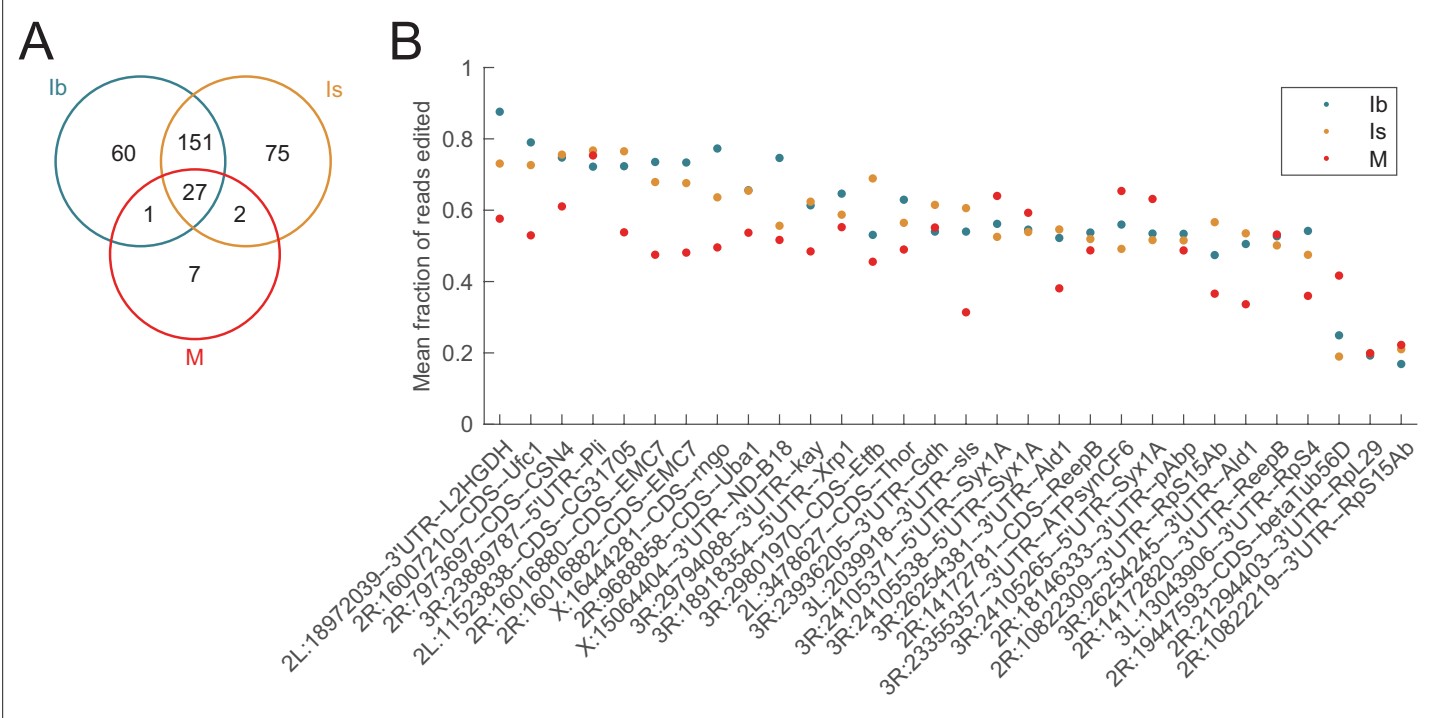

**Figure 6.** Larval muscles edit fewer targets and have a lower editing rate than motoneurons. (**A**) Muscle cells uniquely edit 7 sites, while 30 sites are also found in Ib or Is neurons. (**B**) Muscles generally had a lower editing rate than Ib or Is neurons at sites edited in both cell types.

neuronal populations. Differential editing within the 3'UTR of several genes could impact expression of their encoded proteins, but neuronal subtype diversity of Ib and Is cells appears to be driven mostly by their unique DEGs rather than differential RNA editing.

## Larval muscles have fewer editing targets and reduced editing rate per site compared to MNs

To compare the editing landscape of MNs with a non-neuronal cell type, Patch-seq data generated from larval abdominal muscles 1 and 4 (*Jetti et al., 2023*), which are postsynaptic targets for Ib and Is MNs (*Figure 1C*), was analyzed using the same pipeline as described above. *Drosophila* body wall muscles are generated from fusion of individual myocytes, forming polynuclear cells containing ~8–20 nuclei that share a common cytoplasm (*Windner et al., 2019*). As such, Patch-seq from larval muscles is likely to sample a larger population of editing possibilities compared to single nuclei MNs. Despite being polynucleated, only 37 canonical editing sites were detected in at least 10 muscle samples that had >10% editing rate (*Supplementary files 1, 2, and 7*, *Figure 6A*). Seven of these sites were uniquely edited in muscles, and the remaining 30 were also present in Ib or Is MNs (*Figure 6A*). Muscle cells displayed similar or lower editing rates than Ib or Is neurons at shared edit sites (*Figure 6B*, *Supplementary file 2*). Muscle-specific editing sites (*Supplementary file 7*) were enriched for highly expressed muscle genes, including *Act5C* (Actin homolog), *wupA* (Troponin 1 homolog), *TpnC47D* (Troponin C homolog), and *mthl3* (Secretin-like GPCR). Like MNs, editing rate in muscles was variable for specific target sites, ranging from 18% at the lower end to 75% edited (*Supplementary file 7*). Compared to an average editing rate per site of 66% for MN targets, the average editing rate for muscle-only targets was 34%. These data indicate larval MNs have more editing sites and higher editing rates than their postsynaptic muscle targets, consistent with RNA editing having a greater impact on neuronal function.

## Noncanonical RNA editing is observed in larval MNs but results in fewer significant amino acid changes

Although canonical A-to-I ADAR-dependent deamination is well-known, other forms of mRNA editing have been described and are emerging in importance due to their application in targeted genome editing (*Christofi and Zaravinos, 2019*; *Xu et al., 2022*; *Zhang et al., 2024*). By comparing MN transcriptomes to parental genome sequences, we identified noncanonical ribonucleoside substitutions (non-A-to-I) across several mRNAs (*Figure 1D*, *Supplementary files 3 and 4*). A similar percent of edits occurred in the CDS for canonical (55%) and noncanonical (48%) editing (*Figures 7A, B and 1E, F*). However, the fraction of silent changes was much higher for noncanonical than canonical edits (70% vs. 42%, *Figure 7C*). Although fewer of these edits resulted in an amino acid change, they generated a distinct pattern of substitutions compared to ADAR-dependent editing (*Figure 7—figure supplement 1A*). Unlike canonical editing, noncanonical edit sites were not enriched for neuron-specific genes (*Supplementary file 4*, *Figure 7—figure supplement 1B*). However, the overall fraction of cells displaying noncanonical RNA edits (*Figure 7—figure supplement 1C*) and the likelihood of edits to cluster near each other (*Figure 7—figure supplement 1D*) were similar to ADAR-dependent editing. In addition, there was evidence for a rate-limiting noncanonical editing machinery for abundantly edited transcripts (*Figure 7D*) and a reduced number of overall edits for cells with a high editing rate (*Figure 7E*). Noncanonical editing also exhibited variability in rate per site across individual cells for both Ib and Is subtypes, with some neurons displaying higher or lower editing rates (*Figure 7F*). Overall, Ib and Is neurons showed similar numbers of edited sites (*Figure 7—figure supplement 1E*), editing rate per individual site (*Figure 7—figure supplement 1F*), and fraction of cells containing a specific noncanonical edit (*Figure 7—figure supplement 1G*).

Only a few noncanonical edits were identified that altered conserved residues in key neuronal proteins (*Supplementary file 4*), suggesting a limited impact on MN function. One edit of interest was identified in the activity-regulated *Arc1* mRNA that encodes a retroviral-like Gag protein involved in synaptic plasticity (*Ashley et al., 2018*; *Hantak et al., 2021*). This C-to-T edit resulted in a P124L amino acid change, with ~50% of *Arc1* mRNA edited at this site for both Ib and Is neurons. Arc1 assembles into higher order viral-like capsids (*Erlendsson et al., 2020*) and P124 resides in the hinge region between the central capsomere structure (amino acids 42–120) and the C terminal end (amino acids 125–203) that protrudes from the core capsomere. P124 is highly conserved in other insect Arc homologs, suggesting the edit could alter capsid formation or structure. We also identified this *Arc1* C-to-T edit in publicly available RNAseq datasets from adult *Drosophila* antennal olfactory neurons (*Harrell et al., 2021*), with an edit frequency ranging from 20–80% (*Figure 7—figure supplement 2A–C*). Although *Arc1* is an activity-regulated gene, an increase in neuronal activity driven by overexpression of the inward rectifier Kir2.1 channel did not impact its editing (*Figure 7—figure supplement 2C*, p = 0.3792, Student's *t*-test, *t* statistic = 0.8, df = 22). The *Arc1* C-to-T edit was also identified in another publicly available RNAseq dataset from adult *Drosophila* brains (*Bukhari et al., 2024*). While the edit was observed at ~50% in control brains, it was absent in a matched Alzheimer's disease model expressing pathogenic Tau proteins (*Figure 7—figure supplement 2D*), suggesting pathogenic Tau expression alters *Arc1* editing. Given *Arc1* mRNA levels are upregulated following Tau expression (*Bukhari et al., 2024*), the higher abundance of the mRNA may become rate-limiting for editing and partially account for the reduction. We also identified a noncanonical G-to-A edit that resulted in a G2E missense mutation in Frequenin 1 (Frq1), a multi-EF hand containing $Ca^{2+}$-binding protein that regulates synaptic transmission (*Dason et al., 2009*). This glycine residue is conserved in all Frq homologs and is the site of *N*-myristoylation, a co-translational lipid modification that tethers Frq proteins to the plasma membrane (*Farazi et al., 2001*). Some Frq family members cycle on and off membranes when this domain is sequestered in the absence of $Ca^{2+}$ (*Burgoyne et al., 2004*). Although the G2E edit is found at lower levels (20% in Ib MNs and 18% in Is MNs), it could create a pool of soluble Frq1 that alters downstream signaling. Given the overall lack of neuronally enriched target sites and a larger propensity for generating silent coding changes, we conclude that the widespread effects of canonical A-to-I editing within *Drosophila* larval MNs likely play a more significant role in generating functional protein diversity than noncanonical editing.

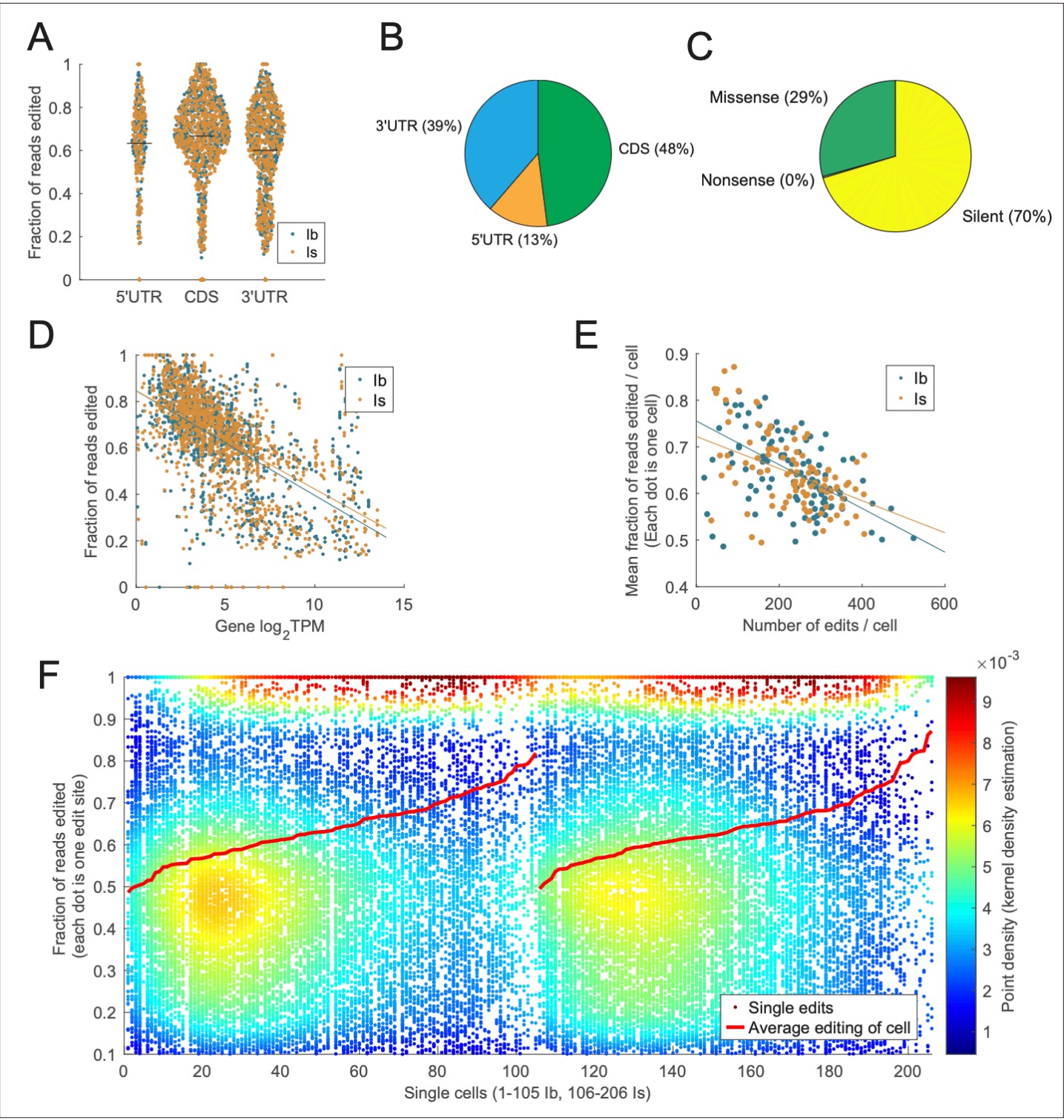

**Figure 7.** Noncanonical editing is not enriched for neuron-specific genes and causes more silent changes than canonical editing. (**A**) The editing rate across 5′UTR, CDS, and 3′UTR was similar to canonical editing. (**B**) The fraction of edits in the 5′UTR, CDS, and 3′UTR was similar to canonical editing. (**C**) The fraction of RNA edits resulting in silent amino acid mutations was much higher for noncanonical editing (70%) than canonical editing (42%). (**D**) Noncanonical editing rate is inversely proportional to the expression level of the edited gene for both Ib and Is neurons. (**E**) Neurons with fewer noncanonical editing sites had a higher mean editing rate. (**F**) Single neurons expressed noncanonical edits with a wide range of editing levels, although mean noncanonical editing rate (red line) was more homogenous than canonical editing.

The online version of this article includes the following figure supplement(s) for figure 7:

*Figure 7 continued on next page*

**eLife** Research article

*Figure 7 continued*

**Figure supplement 1.** Noncanonical editing is similar across Ib and Is neurons and displays variable editing rates and a rate-limiting machinery for abundant mRNAs.

**Figure supplement 2.** Noncanonical *Arc1* editing is not activity regulated but disrupted following mutant Tau overexpression.

## Comparing Patch-seq target sites and editing rate to other larval and adult neuron RNAseq datasets

We next identified RNA editing targets in several other publicly available RNAseq datasets that used pooled larval or adult neurons to compare to those found using our Patch-seq approach. Raw RNAseq datasets were analyzed from three sources: (1) third instar larval MNs that were sorted by FACS (fluorescence-activated cell sorting) using the type I MN-specific *OK6*-GAL4 driver (*Cypranowska et al., 2024*), hereafter referred to as 'Larva'; (2) nuclei from adult head neurons FACS sorted using the pan-neuronal *elav*-GAL4 driver (*Sapiro et al., 2019*), hereafter referred to as 'Adult1'; and (3) total nuclei isolated from adult heads comparing nascent versus polyA+ RNA (*Rodriguez et al., 2012*), hereafter referred to as 'Adult2'. Additional SNP filtering was done by comparing Patch-seq data with genomic sequence from control *yw* adults from the Adult2 study (*Rodriguez et al., 2012*). Sites displaying any reads with the edited base were eliminated from further analysis. After this additional SNP filtering, 127 total canonical CDS, 111 canonical UTR, and 403 noncanonical CDS sites from the Patch-seq data were examined further (*Supplementary files 8 and 9*).

Similar to Patch-seq editing, we required that >10% of the mRNA was edited at an individual site to be considered a positive hit across the three additional datasets. Among 127 Patch-seq canonical CDS edits, 67 were previously annotated in Flybase (*Figure 8A*). As expected, all of the previously annotated sites were identified in at least one other RNAseq dataset, with most being detected in all three groups (*Figure 8B*). Among the 60 unannotated sites found using Patch-seq, 58.3% were validated in at least one other dataset (*Figure 8C, D*). The canonical R-to-G edit in the Rim AZ protein was robustly detected in all three groups and represents an example of an unannotated edit supported in all RNAseq datasets examined. From the three datasets, the largest proportion of unannotated edits was in the Larva RNAseq data, suggesting they may be MN specific or developmentally regulated (*Figure 8D*). These edits include a D-to-G edit in CG42613, a CUB-domain containing membrane protein homologous to the Tolloid-like protein 1 (TLL1) protease, a D-to-G edit in Ankyrin 2 (Ank2), and an E-to-G edit in the Hormone receptor-like 78 (Hr78) protein. Other missense-causing edits in this group included an I-to-V edit in BetaTub97EF and a M-to-V edit in the Shab $K^+$ channel. Unannotated edits in the Larva dataset that were present generally had higher editing rates (>50%) in the Patch-seq data, suggesting some unannotated canonical CDS edits observed only with Patch-seq might be specific to the MN1-Ib and MN-Is neuron subtypes and masked in pooled samples. Several canonical CDS edits from the Patch-seq data were not identified in the other datasets, including the S-to-G edit in Beag, the D-to-G edit in Bsk, and the E-to-G edit in RNaseZ. These edits could be neuron subtype-specific or represent false positives. Further studies will be required to confirm the validity of this smaller subset.

To determine if highly edited sites identified via Patch-seq displayed similarly high rates in other datasets, the editing frequencies from pooled 1b and 1 s MNs from this study were compared to compiled frequencies from the other three RNAseq groups. For previously annotated sites, a positive correlation in editing frequency was observed across all three datasets (Larva vs. Patch-seq annotated: Pearson's $r = 0.78$, $r^2 = 0.61$; Adult1 vs. Patch-seq: Pearson's $r = 0.70$, $r^2 = 0.49$; Adult2 vs. Patch-seq: Pearson's $r = 0.67$, $r^2 = 0.45$, *Figure 8E–G*, blue dots). For example, the edit in nAChRalpha5 resulting in a T791A change was highly edited across all datasets (>90%), while the ATPα sodium pump edit converting E-to-G had lower editing levels (5–20%), consistent with Patch-seq. The correlation plots also identified several sites whose editing frequencies increased in adult brain samples. The R-to-G edit in nAchRbeta1 and the N-to-S edit in Para displayed higher editing levels in the Adult1 (nAChR: 63%, para: 80%) and Adult2 (nAChR: 66%, para: 82%) datasets relative to Patch-seq (nAChR: 36%, para: 68%) and Larva (nAChR: 16%, para: 48%) data (*Figure 8E–G*). A weaker positive correlation was observed for unannotated sites (Larva vs. Patch-seq unannotated: Pearson's $r = 0.40$, $r^2 = 0.16$; Adult1 vs. Patch-seq: Pearson's $r = 0.47$, $r^2 = 0.22$; Adult2 vs. Patch-seq: Pearson's $r = 0.27$, $r^2 = 0.07$, *Figure 8E–G*, orange dots). This observation suggests some unannotated sites are likely Ib and Is

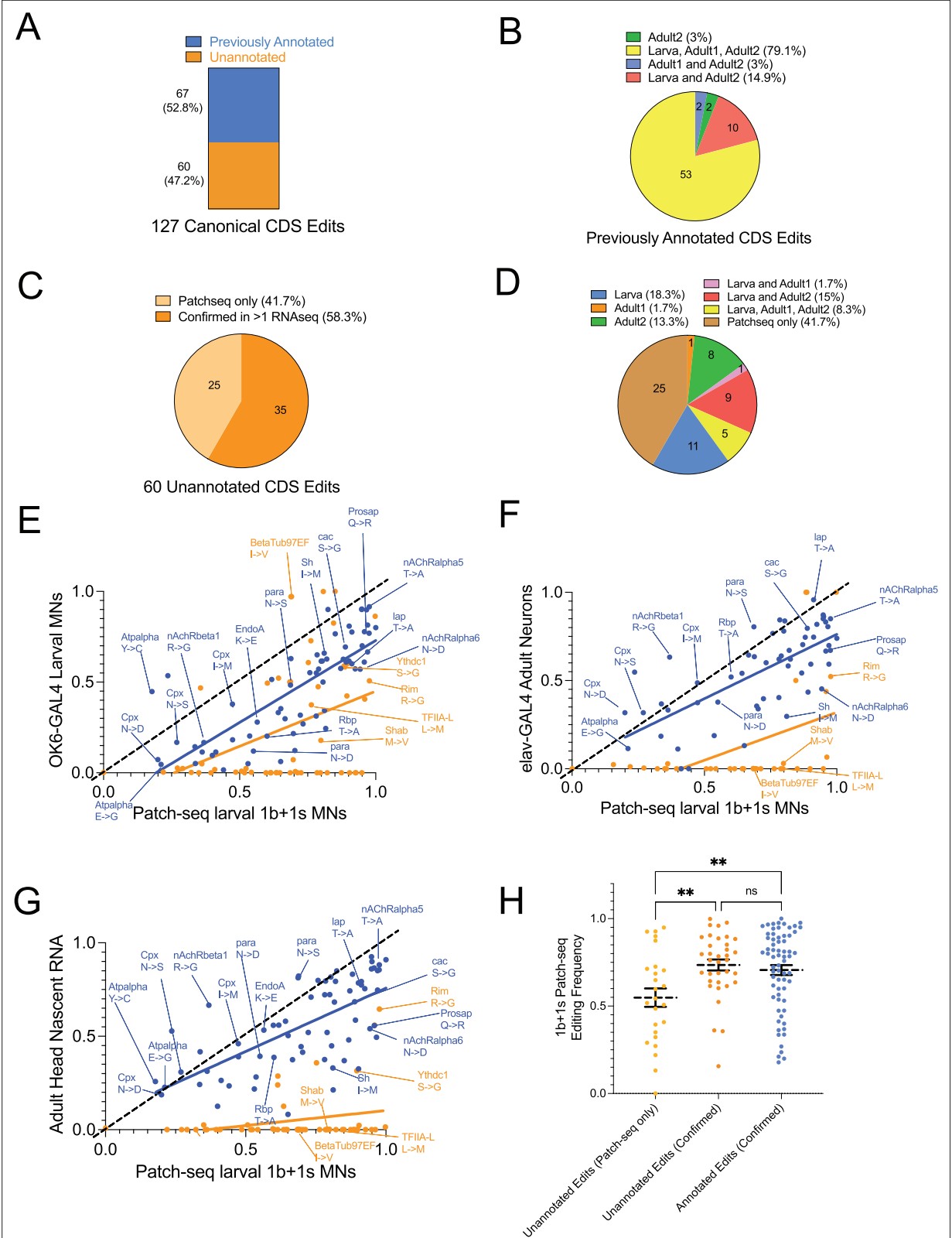

**Figure 8.** Comparison of Patch-seq RNA edits with other RNAseq datasets. (**A**) Number of Patch-seq canonical CDS edits that have been previously annotated on Flybase (blue) versus unannotated ones (orange). (**B**) Previously annotated canonical Patch-seq CDS edits mapped onto other RNAseq datasets. (**C**) For unannotated canonical CDS edits, the number of confirmed edits from other RNAseq datasets (dark orange) versus those only identified in Patch-seq (light orange). (**D**) Unannotated Patch-seq CDS edits mapped onto other RNAseq datasets. Correlation plots of editing

*Figure 8 continued on next page*

*Figure 8 continued*

frequencies for canonical CDS sites between Patch-seq (pooled average of Ib and Is motoneurons [MNs]) and (**E**) six pooled replicates from larval MNs FACS sorted using *OK6*-GAL4 (***Cypranowska et al., 2024***), (**F**) three pooled replicates from nuclei of adult head neurons FACS sorted using *elav*-GAL4 (***Sapiro et al., 2019***), and (**G**) four pooled replicates of nascent RNA from adult heads (***Rodriguez et al., 2012***). Blue dots indicate previously annotated sites and orange dots indicate unannotated sites. Dashed black lines represent a reference for perfect positive correlation, solid blue lines represent actual linear regression lines for annotated sites, and solid orange lines are for unannotated sites. Select edits causing an amino acid change are labeled. Edit sites with no reads in the RNAseq datasets are not shown. (**H**) Scatterplot of Patch-seq editing frequencies for canonical CDS sites in unannotated edits identified only in Patch-seq (light orange, first column), unannotated edits confirmed in other RNAseq (dark orange, second column), and annotated edits confirmed in other RNAseq (blue, third column). Black lines with error bars represent mean ± SEM. Asterisks denote *p* values of: **≤0.01.

The online version of this article includes the following figure supplement(s) for figure 8:

**Figure supplement 1.** Comparison of Patch-seq canonical UTR edits with other RNAseq datasets.

**Figure supplement 2.** Comparison of Patch-seq noncanonical CDS edits with other RNAseq datasets.

**Figure supplement 3.** ADAR dependency, co-transcriptional editing, and the effects of activity reduction.

MN-specific, and the editing rate becomes diluted in pooled RNAseq datasets, while others may represent false positives. The Patch-seq editing frequencies for unannotated sites (0.74) and annotated sites (0.71) confirmed in at least one dataset were significantly higher compared to unannotated edits identified only with Patch-seq (0.55, ANOVA *F* statistic = 5.732, df = 2, Tukey p = 0.0042), indicating the higher the editing rate found in Patch-seq, the greater the likelihood the edit was detected in other published studies (***Figure 8H***).

Among the 111 Patch-seq canonical UTR edits, 16 were previously annotated in Flybase and 95 were unannotated (***Figure 8—figure supplement 1A***). As with CDS sites, 100% of the previously annotated sites were validated, with the majority being detected in all three groups (***Figure 8—figure supplement 1B***). Among the 95 unannotated sites, 63.2% were validated in at least one dataset (***Figure 8—figure supplement 1C***). Of these confirmed unannotated edits, 14.7% were detected only in the Larva dataset (***Figure 8—figure supplement 1D***). Three Syx 5′UTR edits belonged to this category, indicating these represent larval and/or MN-specific edits. Eleven of the twelve unannotated Cpx 3′U′TR edits were also validated across the other datasets. In terms of UTR editing frequency, there was a positive correlation between Patch-seq and other RNAseq data for previously annotated sites (Larva vs. Patch-seq annotated sites: Pearson's $r = 0.90$, $r^2 = 0.81$; Adult1 vs. Patch-seq annotated sites: Pearson's $r = 0.90$, $r^2 = 0.81$; Adult2 vs. Patch-seq annotated sites: Pearson's $r = 0.84$, $r^2 = 0.71$, ***Figure 8—figure supplement 1E–G***, blue dots), indicating the relative editing levels at these sites are robust and static across development and neuronal cell types. For example, the previously annotated Slo 3′UTR 3R:24707423 site was highly edited (80–96%) across all three RNAseq groups and correlated with Patch-seq levels (94%). Similarly, the previously annotated EndoA 3′UTR 3R:18910153 site was edited at moderate levels across the three groups (44–57%) and the Patch-seq data (66%). Similar to CDS edits, editing levels for unannotated UTR sites showed weaker correlation (Larva vs. Patch-seq unannotated sites: Pearson's $r = 0.22$, $r^2 = 0.047$; Adult1 vs. Patch-seq unannotated sites: Pearson's $r = 0.15$, $r^2 = 0.02$; Adult2 vs. Patch-seq unannotated sites: Pearson's $r = 0.09$, $r^2 = 0.008$, ***Figure 8—figure supplement 1E–G***, orange dots). From this group, several sites displayed consistent editing frequencies, including an edit in the Sh 3′UTR (30–50% across the three groups and 67% in Patch-seq) and an edit in the nAchRbeta1 3′UTR (30–70% across the three groups and 50% in Patch-seq). Unlike CDS sites, the Patch-seq editing frequencies for unannotated UTR sites (0.57) and annotated sites (0.51) confirmed by at least one dataset were similar to sites only identified in Patch-seq (0.57, ANOVA *F* statistic = 0.4950, df = 2, Tukey p = 0.6110, ***Figure 8—figure supplement 1H***), indicating UTR edits are generally not unique to Ib and Is MN cell types.

All 403 noncanonical edits identified in Patch-seq were previously unannotated, as most editing studies have excluded non-A-to-I editing from analysis. Among the noncanonical edits, 45% (182/403) were confirmed in at least one of the RNAseq datasets (***Figure 8—figure supplement 2A***). The largest proportion of confirmed edits (16.4%) was verified in the Adult2 group, with 13.7% of confirmed edits verified only in the Larva dataset (***Figure 8—figure supplement 2B***). Noncanonical edits that appeared only in the Patch-seq and Larva datasets included an E-to-Q edit in the RNA-binding protein Staufen, a G-to-E edit in the mitochondrial ribosomal protein mRpS5, and an A-to-S edit in the ribosomal protein RpS15. Noncanonical edits in this category that are predicted to be less disruptive

included A-to-V and E-to-D edits in the ribosomal protein RpS15Ab and a M-to-L edit in Syx1A. Editing frequencies for noncanonical CDS events between Patch-seq and other RNAseq datasets were poorly correlated (Larva vs. Patch-seq: Pearson's $r$ = 0.25, $r^2$ = 0.065; Adult1 vs. Patch-seq: Pearson's $r$ = 0.21, $r^2$ = 0.045; Adult2 vs. Patch-seq: Pearson's $r$ = 0.11, $r^2$ = 0.012, **Figure 8—figure supplement 2C–E**). Similar to canonical CDS edits, noncanonical edits that were more highly edited (>50%) in Patch-seq data were more likely to be detected in other datasets. Indeed, Patch-seq editing frequencies of noncanonical sites confirmed in other datasets were significantly higher (0.68) than sites identified only in Patch-seq (0.51), indicating the higher the editing in larval MNs, the more likely the edit was detected in other datasets (**Figure 8—figure supplement 2F**, p < 0.0001, Student's $t$-test, $t$ statistic = 7.304, df = 368). Several noncanonical editing events also clustered at 50% or 100% editing in the Larva and Adult1 datasets, raising the possibility they correspond to previously unknown SNPs from the parental genotypes used in these studies. Indeed, this clustering was not observed in the Adult2 dataset where parental genomic DNA was sequenced and all SNPs removed. We conclude that although noncanonical editing events are found in all three datasets, the lower correlation with the Patch-seq data suggests some of these sites may represent unknown SNPs or occur at higher frequency in larval MNs.

To examine if editing identified by Patch-seq was disrupted in *Adar* mutants, editing frequency was compared between adult head samples from control and *Adar* null mutant datasets (**Rodriguez et al., 2012**). The vast majority of canonical CDS and UTR editing events identified, including previously unannotated sites, were abolished in *Adar* mutants (**Figure 8—figure supplement 3A, B**, **Supplementary file 8**), indicating these are ADAR-dependent. Noncanonical CDS sites were largely unaffected by loss of ADAR as expected (**Figure 8—figure supplement 3C**, **Supplementary file 9**). To examine if edits identified by Patch-seq were likely to occur co-transcriptionally, editing frequencies between nascent RNA (isolated from adult head nuclei) and polyA+ RNA (isolated from adult heads) were compared using datasets from **Rodriguez et al., 2012**. Editing levels for canonical CDS and UTR edits were well correlated between nascent and polyA+ mRNA (canonical CDS: Pearson's $r$ = 0.90, $r^2$ = 0.80; canonical UTR: Pearson's $r$ = 0.88, $r^2$ = 0.78, **Figure 8—figure supplement 3D, E**), suggesting most edits occur co-transcriptionally, consistent with previous findings (**Rodriguez et al., 2012**). In contrast, noncanonical editing sites were poorly correlated between nascent and polyA+ mRNA (Pearson's $r$ = 0.34, $r^2$ = 0.12, **Figure 8—figure supplement 3F**). A subset of these sites had higher editing in polyA+ mRNA (i.e. R-to-H in CG42518, 48% edited in polyA+ vs. 0% in nascent RNA), suggesting some noncanonical editing may preferentially occur in processed mRNA following nuclear export. Indeed, proteins implicated in C-to-U and G-to-A editing localize to both the nucleus and cytoplasm in mammalian cells (**Anant et al., 2001**; **Ikeda et al., 2011**).

Finally, we examined if any RNA editing events identified by Patch-seq were activity-regulated by taking advantage of Larva RNAseq data obtained following reductions in neurotransmitter release using knockdown of the AZ proteins RBP and Unc13 (**Cypranowska et al., 2024**). Disruptions to neurotransmitter release under these conditions result in a homeostatic response in which MNs increase their excitability to compensate. Given many editing targets in MNs regulate neuronal excitability or synaptic transmission, changes in editing at these sites might alter MN excitability. Editing frequencies were compared for canonical CDS, canonical UTR, and noncanonical CDS sites in larval MN RNAseq datasets from controls or following RBP or Unc13 RNAi (**Figure 8—figure supplement 3G–L**). Overall, editing frequency was largely unchanged following presynaptic disruptions to neurotransmitter release (RBP vs. control canonical CDS: Pearson's $r$ = 0.94, $r^2$ = 0.88; RBP vs. control canonical UTR: Pearson's $r$ = 0.96, $r^2$ = 0.91; RBP vs. control noncanonical CDS: Pearson's $r$ = 0.96, $r^2$ = 0.93; Unc13 vs. control canonical CDS: Pearson's $r$ = 0.94, $r^2$ = 0.87; Unc13 vs. control canonical UTR: Pearson's $r$ = 0.97, $r^2$ = 0.94; Unc13 vs. control noncanonical CDS: Pearson's $r$ = 0.96, $r^2$ = 0.93, **Figure 8—figure supplement 3G–L**). However, a few sites displayed evidence for a change in editing rate. Among canonical CDS sites, the Y-to-C edit in the ATPα sodium pump was markedly reduced for both RNAi conditions (RBP: 6%, Unc13: 9%) compared to controls (44%). Editing rates for the Shab S-to-G and Cac S-to-G sites were also reduced following Unc13 knockdown. A modest increase in the Rim R-to-G edit was observed in both RBP and Unc13 RNAi knockdown datasets. For UTR editing, a slight increase in the nAchRbeta1 3′UTR site and a modest decrease in the Slo 3′UTR editing frequencies were observed in both RNAi datasets. A 3′UTR edit in Sh was slightly increased following Rbp RNAi. Several noncanonical edits were also affected by presynaptic weakening (**Figure 8—figure**

*supplement 3I, L*), including a decrease in an M-to-L edit in AnxB10 in both knockdown conditions. Given Rim, Cac, Shab, Sh, and Slo mRNAs were identified as downregulated DEGs under these hypo-activity conditions (*Cypranowska et al., 2024*), it will be informative to examine if these changes in target editing frequency contribute to their reduced mRNA expression.

## Discussion

Multiple types of ribonucleoside modifications occur across mRNAs, tRNAs, rRNAs, and ncRNAs (*McCown et al., 2020*). RNA editing modifies single nucleotides in mRNA transcripts in a cell-type-specific manner and is one of the more widely observed RNA modifications across metazoans. In this study, we characterized the RNA editing landscape of individual *Drosophila* third instar larval MN populations using Patch-seq RNA extraction to define rules for single-cell editing and identify the most relevant editing targets. Tonic-like Ib and phasic-like Is MNs form well-studied glutamatergic synapses with cell-type-specific electrophysiological and morphological features, providing a model system to investigate how RNA editing contributes to their morphology and function, as well as any role in cell-type-specific diversity. The majority of A-to-I edits identified in our study occurred within mRNA coding regions, with 60 edits predicted to cause significant alterations to protein function. As observed in prior studies (*Hoopengardner et al., 2003*; *Jepson and Reenan, 2009*; *Palladino et al., 2000a*; *Rueter et al., 1999*; *Sapiro et al., 2019*), mRNAs encoding ion channels and synaptic proteins were among the most highly edited targets, suggesting editing of these gene classes is likely to fine-tune membrane excitability and synaptic output in larval MNs. The overall RNA editing landscape was similar across Ib and Is populations, indicating differences in editing targets or editing rate is unlikely to be a major driver of cell-type diversity for these neuronal subtypes.

Using single-cell sequencing from stereotyped and identified single Ib and Is MNs, this analysis revealed a wide variation in average RNA editing levels for specific sites and across individual cells. Indeed, many sites underwent highly stochastic editing when compared across the neuronal population. For example, some neurons displayed ~100% editing at certain sites, while others displayed no editing for the same target. Such dramatic differences in editing rate at specific target sites are likely to contribute to the heterogeneous features observed within the same neuronal population. At the single neuron level, each cell would be predicted to express a population of both edited and unedited proteins. What drives the variability in RNA editing across individual MNs is unclear. Given that ADAR is an enzyme, its affinity for specific dsRNA structures plays a role in establishing editing levels for any specific pre-mRNA target site. Although *Adar* mRNA level in individual neurons was not correlated with editing rate, the expression level of the target mRNA undergoing editing was a major factor. Indeed, mRNA expression level was anti-correlated with editing percent, suggesting *Drosophila* ADAR activity becomes rate-limiting for abundant pre-mRNAs as proposed in mammals (*Lopdell et al., 2019*). Beyond mRNA expression level, variable expression of other trans-acting factors that regulate ADAR function or pre-mRNA accessibility could modulate editing rate or target site selection across individual neurons as previously hypothesized (*Ingleby et al., 2009*). Given single-cell DNA–RNA mismatches were excluded if fewer than 10 total RNAseq reads covered the site, editing calls for poorly expressed mRNAs within these neuronal populations are not included in this study. We also identified edits that were only present in larval MNs (this study and *Cypranowska et al., 2024*) and not observed in adult RNAseq datasets, consistent with cell type or developmental stage-specific editing. In terms of activity regulation, a few targets showed changes in editing frequency when presynaptic neurotransmitter release was reduced.

Although this study defines the RNA editing landscape for *Drosophila* larval MNs, future experiments will be required to elucidate how specific RNA editing events regulate target gene expression or protein function. Of particular interest will be edits occurring at high levels, such as the 3'UTR edit in *ATPsynCF6* and 5'UTR edits in *Syx1A*, as these may represent edits required for normal transcript localization, expression, or function. For editing events that lead to amino acid change, these substitutions could in principle improve, disrupt, or modify protein function. We failed to observe cases of re-coding an amino acid from a non-conserved residue into a residue conserved in orthologous proteins, suggesting most coding edits with a functional effect are likely to diversify or disrupt protein activity as previously suggested based on comparative genomics (*Duan et al., 2023*; *Liu et al., 2024*). The evolutionary advantage, if any, that RNA editing has in modifying protein function is still unclear. One hypothesis is that RNA editing plays similar roles to alternative splicing. For example, changes

in the ratio of Unc13A and Unc13B splice isoforms during *Drosophila* AZ development regulate the probability of SV release by altering the position of docked SVs near $Ca^{2+}$ channels (***Pooryasin et al., 2021***; ***Reddy-Alla et al., 2017***). In a similar manner, ratiometric pools of edited versus unedited proteins could provide a mechanism to fine-tune specific neuronal features, particularly for proteins whose abundance is rate-limiting or that act within multimeric complexes. Such a mechanism has been observed for distinct Cpx edit variants (***Brija et al., 2023***), where individual Cpx proteins are required to clamp multiple assembling SNAREpins during SV priming. In addition, functional properties of the Shaker voltage-gated $K^+$ channel are shaped by the combination of individual edits across 4 distinct sites (***Ingleby et al., 2009***).

Canonical and noncanonical (non-A-to-I) editing was observed in larval MNs, though ADAR-dependent editing resulted in far more edits to neuronal-specific genes predicted to impact protein function. The majority of noncanonical CDS edits caused silent changes, suggesting evolutionary selection against this mechanism as a pathway for generating protein diversity. One noncanonical C-to-U coding edit of interest caused a P124L amino acid change in the activity-regulated *Arc1* mRNA that encodes a retroviral-like Gag protein involved in synaptic plasticity (***Hantak et al., 2021***). Approximately 50% of total *Arc1* mRNA was edited at this site in both Ib and Is neurons. Given *Drosophila* Arc1 assembles into higher order viral-like capsids similar to mammalian Arc proteins (***Erlendsson et al., 2020***), it will be interesting to determine if and how this amino acid change might alter capsid formation or structure. In terms of the underlying editing machinery, the mammalian APOBEC cytosine deaminases generate C-to-U edits in target mRNAs like *apoB* (***Salter et al., 2016***). The *Drosophila* genome encodes three predicted cytosine deaminases (CG8349, CG8353, and CG8360), though their role in editing has not been studied. Our Patch-seq data indicate CG8353 and CG8360 are expressed in Ib and Is MNs at similar or higher levels as *Adar* (Ib TPM (transcripts per million): *Adar* – 1.51; CG8353 – 1.53; CG8360 – 4.77; Is TPM: *Adar* – 2.16; CG8353 – 1.44; CG8360 – 4.36), suggesting C-to-U editing may be quite robust in these neuronal subtypes. However, given the large number of silent changes to protein sequence, it seems unlikely that noncanonical editing plays a major role in regulating larval MN function compared to ADAR-dependent editing.

## Materials and methods
### Single neuron Patch-seq RNA sequencing
RNA sequencing data from 105 MN1-Ib and 101 MNISN-Is *Drosophila melanogaster* third instar larval MNs (GEO accession #GSE222976) (***Jetti et al., 2023***) was used for editing analysis. 10XUAS-mCD8-GFP (RRID:BDSC_32185) was expressed using either a Ib-specific (RRID:BDSC_40701) or Is-specific (RRID:BDSC_49227) Gal4 driver. The nucleus and surrounding cytoplasm of GFP+ MNs were extracted using a patch pipet and immediately frozen on dry ice. cDNA was generated, purified, dual indexed, and run on an Illumina NextSeq500 (paired-end 75 nt reads) sequencer as described (***Jetti et al., 2023***).

### Genome sequencing
High purity genomic DNA from parental Ib-Gal4, Is-Gal4, and 10XUAS-mCD8-GFP lines was collected and sequenced. Flies were grown at 25°C. 10 adult male flies from each genotype were collected under $CO_2$ anesthetization into a 1.5-ml centrifuge tube and immediately frozen in liquid nitrogen. 120 µl of grinding buffer (0.2 M sucrose, 0.1 M Tris pH 9.2, 50 mM EDTA, 0.5% SDS) was added and a VWR pellet mixer (VWR, cat# 47747-370) was used to homogenize the flies for 1 min. Samples were vortexed for 10 s and placed on ice for 15 min. 500 µl of phenol:chloroform:isoamyl alcohol (Millipore-Sigma, SKU# 6805-100ML, Omnipur 25:24:1 and TE Buffered Saturated pH 6.7/8.0) was added and vortexed for 30 s. Samples were centrifuged at 16,100 relative centrifugal force (RCF) for 5 min to pellet debris and separate protein to the interphase and DNA in the upper phase. The DNA-containing upper phase was pipetted into 1 ml of 100% ethanol in a new 1.5 ml tube and vortexed briefly to mix. After incubation overnight at 4°C, samples were centrifuged at 16,100 RCF for 10 min to pellet DNA. The pellet was washed in 70% ethanol by pipetting, centrifuged at 16,100 RCF for 10 min, and the pellet air-dried for 30 min. 50 µl of DNase-free ultrapure water was added to each sample and incubated at 37°C for 15 min before resuspension. DNA samples were prepared

as libraries for sequencing using the Nextera DNA Flex (Illumina) kit and sequenced on an Illumina NextSeq500.

## RNA edit detection and parental DNA variation calling

Paired-end fastq files were mapped to *Drosophila_* BDGP6.92 reference using BWA. Sequencing reads with mapping quality score less than 5 were removed from downstream analyses using Samtools. PCR duplicates were removed using GATK MarkDuplicatesSpark. The variations were first called using Samtools mpileup, and the high-quality SNPs (depth ≥8 and quality score ≥20) and indels were selected as the database of known polymorphic sites for GATK base quality recalibration. Base quality recalibration was accomplished using GATK BaseRecalibrator and ApplyBQSR. Variations were then called by GATK HaplotypeCaller with VCF outputs. Variations were subset to exonic regions only. Only SNPs were kept for downstream filtering. Since genes localize to both DNA strands while RNA sequencing reads are only aligned to the positive strand, A-to-G editing in minus strand genes appears as T-to-C. To facilitate data analysis, edits in minus strand genes are listed as their reverse complement in the text.

## Filtering RNA variations against parental DNA SNPs

SNPs in RNA samples were excluded if they existed in any of the three parental DNA samples as determined by GATK HaplotypeCaller and Samtools mpileup based on genomic location and genotype. To prevent mapping complications of nearby indels, RNA SNPs with indels within 10 bp were filtered out. The remaining RNA SNPs were selected as candidate RNA editing sites for further validation.

## Filtering of candidate RNA edits

A custom MATLAB code was used to further analyze and filter variant call output from Samtools and GATK. Potential edits were removed if (1) there were ≤10 RNA sequencing reads covering this genomic location, (2) the GATK Phred-scaled probability (QUAL score) of a polymorphism was ≤20, (3) the cell-specific edit site had ≤10% editing rate, (4) the edit was found in fewer than 10 Ib or Is cells, or (5) less than 96% of the sequenced genomic DNA nucleotide at this site failed to match the reference genome (a genomic SNP). Remaining edits were processed to categorize where in the gene the edit occurred. On rare occasions, edits occurred in overlapping genes. In these cases, only edits occurring within exons were kept and listed as an edit in each gene if more than one gene had an exon at this location. Manual inspection revealed many noncanonical edits/SNPs clustered near the beginning of the 5′UTR (where sequencing errors were more common), so edits <20 nucleotides from the 5′UTR start site were excluded. The remaining high-confidence edit sites were analyzed to provide summary statistics, including average editing rate across Ib and Is neurons. Gene boundaries for CDS, 5′UTR, and 3′UTR, previously annotated A-to-I editing events, gene descriptions, Pfam locations, and other genome-related information were downloaded from FlyBase on September 15, 2023.

### Software and code

Software versions were as follows: Star (2.5.3a), BWA (0.7.12), Samtools (1.10), GATK (4.1.2.0), and MATLAB (2022b). Three-dimensional protein structures were produced using AlphaFold2 (*Jumper et al., 2021*). The full MATLAB code with instructions for RNA editing analysis is downloadable as a ZIP file from Supplemental Material as *Source code 1*.

### RNA editing conservation analysis

Canonical RNA edits that led to a missense change in the translated amino acid were analyzed to determine if the RNA editing site changed an amino acid that was conserved across species. NCBI BLAST was used to locate the relevant amino acid in polypeptide strings of other insects or mammals and compared to the *Drosophila* reference genome. Sites were considered conserved if the unedited amino acid at this location was identical between *Drosophila* and other species.

### Comparison of Patch-seq edits with publicly available RNAseq datasets

Raw fastq files were downloaded from existing *Drosophila* RNA sequencing datasets GEO Accession ID GSE250462: GSM7979414-7979425 (*Cypranowska et al., 2024*); GSE37232: GSM914102,

914107, 914108, 914113, 915213–915215, 915220, 914114–914119, 914095 (*Rodriguez et al., 2012*); GSE113663: GSM3110802–3110804 (*Sapiro et al., 2019*); GSE223345: GSM6945880–6945885 (*Bukhari et al., 2024*); EMBL-EBI ArrayExpress E-MTAB-10065: Samples 13–25 EH1_1, EH1_2, EH1_2.2, EH1_3, EH1_3.2, EH1_4, EH1_5, EH1_6, EH1_7, EH1_8, EH1_9, EH1_10, EH2_1, only 'Lmerged' files (*Harrell et al., 2021*) and mapped to *Drosophila_melanogaster*.BDGP6.32.108 reference genome using STAR (2.7.11b). Neither FastqQC nor adapter trimming steps were performed. The −−outSAMtype BAM SortedByCoordinate command was used to output BAM files sorted by coordinate following alignment. The output files 'Aligned.sortedByCoord.out.bam' were then indexed by running igvtools within Integrative Genomics Viewer IGV, version 2.15.2, (*Robinson et al., 2011*) for visualization. For each genomic dataset loaded into IGV, the genomic coordinate (edit position) for each canonical CDS, canonical UTR, and noncanonical CDS editing event from Patch-seq was inputted. The number of reads with the edit variant (ALT) and those with the unedited variant (REF) were recorded. Editing frequency was calculated as ALT/(REF + ALT).

Patch-seq edit positions were also checked in the gDNA_Fly_yw dataset GSM914095 from GSE37232 (*Rodriguez et al., 2012*). Edit sites that did not show 100% unedited 'REF' reads were eliminated to filter out any additional SNPs. Editing frequencies for the remaining sites were pooled to obtain compiled averages as described below. Editing frequencies recorded for six *OK6*-GAL4>mCherry-NLS replicates (GSM7979414-7979419, from GSE250462) were averaged to obtain the compiled 'Larval' data. Editing frequencies for three *OK6*-GAL4>mCherry-NLS, RBP-RNAi datasets (GSM7979420–7979422) and three *OK6*-GAL4>mCherry-NLS, Unc13-RNAi datasets (GSM7979423–7979425) were averaged to obtain RBP and Unc13 knockdown data. Three UAS-*unc84*-2XGFP; *elav*-GAL4 datasets (GSM3110802–3110804) were averaged to obtain the compiled 'Adult1' data. pAseq_Fly_ZT2_yw (GSM915213), pAseq_Fly_ZT14_yw (GSM915214), pAseq_Fly_ZT0_PE_CS (GM915215), and pAseq_Fly_ZT20_PE_CS (GSM915220) were averaged to obtain the compiled 'Adult2' polyA mRNA data. Nascentseq_Fly_ZT2_R1_yw (GSM914102), Nascentseq_Fly_ZT22_R1_yw (GSM914107), Nascentseq_Fly_ZT2_R2_yw (GSM914108), and Nascentseq_Fly_ZT22_R2_yw (GSM914113) were averaged to obtain the compiled 'Adult2' Nascent mRNA data. Nascentseq_Fly_FM7a_R1 (GSM914116) and Nascentseq_Fly_FM7a_R2 (GSM914117) were averaged to obtain the compiled 'Adult2' control data for comparison to the *Adar* null mutants averaged Nascentseq_Fly_ADAR0_R1 (GSM914118) and Nascentseq_Fly_ADAR0_R2 (GSM914119). These compiled averages were used for generating all correlation plots. For all correlation plots, simple linear regression was performed using GraphPad Prism version 10.5.0 (673) for macOS (GraphPad Software, Boston, Massachusetts USA, https://www.graphpad.com/). In all cases, regression lines were not constrained; the lines were not forced to pass through the origin (0,0).

For *Supplementary files 8 and 9*, each site with a compiled average of at least 10% editing in one of the three datasets was listed as confirmed. Any site with less than 10% editing was listed as not confirmed. Edits with no read coverage were listed as no reads. The number of canonical CDS, canonical UTR, and noncanonical CDS edits belonging to each dataset (Larva, Adult1, or Adult2) was determined and plotted to generate pie charts.

## Statistical analysis

For comparisons between two groups, statistical significance was determined using an unpaired, two-tailed, two-sample Student's *t*-test. For comparisons between three or more groups, statistical significance was determined using a one-way ANOVA followed by the Tukey–Kramer post hoc multiple comparisons test. Figures depict the mean of each distribution and individual data points, and asterisks denote p values of: *$p \leq 0.05$; **$p \leq 0.01$; ***$p \leq 0.001$; and ****$p \leq 0.0001$. Normalcy of data distribution was determined using the one-sample Kolmogorov–Smirnov test. The correlation coefficient (*r*) between two sets of data was determined using the Pearson correlation method.

## Acknowledgements

This work was supported by NIH grants MH104536 and NS117588 to JTL. ABC was supported in part by NIH pre-doctoral training grant F31NS118948. We thank the Bloomington *Drosophila* Stock Center (NIH P40OD018537) for providing *Drosophila* strains, Duanduan Ma and Charlie Whittaker for help with genomic and editing analysis, and members of the Littleton lab for helpful discussions and comments on the project and manuscript.

# Additional information

## Funding

| Funder | Grant reference number | Author |
|---|---|---|
| National Institute of Neurological Disorders and Stroke | NS117588 | J Troy Littleton |
| National Institute of Mental Health | MH104536 | J Troy Littleton |
| National Institute of Neurological Disorders and Stroke | F31NS118948 | Andrés B Crane |

The funders had no role in study design, data collection, and interpretation, or the decision to submit the work for publication.

## Author contributions

Andrés B Crane, Conceptualization, Formal analysis, Investigation, Methodology, Writing – original draft, Writing – review and editing; Michiko O Inouye, Formal analysis, Investigation, Writing – review and editing; Suresh K Jetti, Investigation, Writing – review and editing; J Troy Littleton, Conceptualization, Supervision, Funding acquisition, Writing – original draft, Project administration, Writing – review and editing

## Author ORCIDs

Andrés B Crane ⓘ https://orcid.org/0000-0002-0242-526X
J Troy Littleton ⓘ https://orcid.org/0000-0001-5576-2887

Reviewer #1 (Public review): https://doi.org/10.7554/eLife.108282.2.sa1
Reviewer #2 (Public review): https://doi.org/10.7554/eLife.108282.2.sa2
Reviewer #3 (Public review): https://doi.org/10.7554/eLife.108282.2.sa3
Author response https://doi.org/10.7554/eLife.108282.2.sa4

# Additional files

## Supplementary files

Supplementary file 1. Canonical RNA (A-to-I) editing sites identified across all individual Ib motoneurons (MNs), Is MNs, and abdominal muscles of third instar larvae.

Supplementary file 2. Summary of canonical RNA editing sites identified in Ib motoneurons (MNs), Is MNs, and abdominal muscles.

Supplementary file 3. Noncanonical RNA (non-A-to-I) editing sites identified across all individual Ib motoneurons (MNs), Is MNs, and abdominal muscles of third instar larvae.

Supplementary file 4. Summary of noncanonical RNA editing sites identified in Ib motoneurons (MNs), Is MNs, and abdominal muscles.

Supplementary file 5. Canonical RNA editing targets with the highest editing rate that cause amino acid substitutions.

Supplementary file 6. Canonical RNA editing site differences in Ib versus Is motoneurons (MNs) following removal of differentially expressed genes (DEGs).

Supplementary file 7. Canonical RNA editing sites identified in larval abdominal muscles.

Supplementary file 8. Comparison of canonical RNA editing sites identified via Patch-seq to other RNAseq datasets.

Supplementary file 9. Comparison of noncanonical RNA editing sites identified via Patch-seq to other RNAseq datasets.

MDAR checklist

Source code 1. Zip file of MATLAB code for RNA editing detection. This file contains the necessary

MATLAB codes to process output of GATK/SAMtools variant callers as described in methods.

## Data availability

The source data supporting this study are available within the paper, supplementary material, and indicated repositories. RNA sequencing data is available at the GEO data repository (accession #GSE222976). DNA sequencing data for parental genotypes is available at the NCBI BioSample Database and Sequence Read Archive (BioProject #PRJNA1112347). Existing RNA sequencing datasets used for comparison to Patch-seq edits were accessed through GEO and EMBL-EBI ArrayExpress public repositories: GSE250462 (Cypranowska et al., 2024), GSE37232 (Rodriguez et al., 2012), GSE113663 (Sapiro et al., 2019), GSE223345 (Bukhari et al., 2024), E-MTAB-10065 (Harrell et al., 2021). MATLAB codes are provided in Source code 1.

The following dataset was generated:

| Author(s) | Year | Dataset title | Dataset URL | Database and Identifier |
| --- | --- | --- | --- | --- |
| Crane AB, Inouye MO, Jetti SK, Littleton JT | 2024 | A stochastic RNA editing process targets a limited number of sites in individual Drosophila glutamatergic motoneurons | https://www.ncbi.nlm.nih.gov/bioproject/PRJNA1112347 | NCBI BioProject, PRJNA1112347 |

The following previously published datasets were used:

| Author(s) | Year | Dataset title | Dataset URL | Database and Identifier |
| --- | --- | --- | --- | --- |
| Jetti SK, Crane AB, Akbergenova Y, Aponte-Santiago NA, Cunningham KL, Whittaker CA, Littleton JT | 2023 | Molecular Logic of Synaptic Diversity Between Drosophila Tonic and Phasic Motoneurons | https://www.ncbi.nlm.nih.gov/gds/?term=GSE222976 | NCBI Gene Expression Omnibus, GSE222976 |
| Cypranowska C, Isacoff E | 2024 | Effect of RNAi-mediated knockdown of unc-13 and Rbp on gene expression in motor neurons of *Drosophila melanogaster* third instar larva | http://www.ncbi.nlm.nih.gov/geo/query/acc.cgi?acc=GSE250462 | NCBI Gene Expression Omnibus, GSE250462 |
| Rodriguez J, Menet JS, Rosbash M | 2012 | Nascent-seq indicates widespread cotranscriptional RNA editing in Drosophila | https://www.ncbi.nlm.nih.gov/geo/query/acc.cgi?acc=GSE37232 | NCBI Gene Expression Omnibus, GSE37232 |
| Sapiro AL, Shmueli A, Henry GL, Li Q | 2019 | A-to-I RNA editing signatures of neuronal populations within the Drosophila brain | https://www.ncbi.nlm.nih.gov/geo/query/acc.cgi?acc=GSE113663 | NCBI Gene Expression Omnibus, GSE113663 |
| Bukhari H, Feany MB | 2024 | Gene expression profile at single cell levels of the cells from the controls and tau P251L Drosophila brain | https://www.ncbi.nlm.nih.gov/geo/query/acc.cgi?acc=GSE223345 | NCBI Gene Expression Omnibus, GSE223345 |
| Harrell et al | 2021 | Changes in presynaptic gene expression during homeostatic compensation at a central synapse | https://www.ebi.ac.uk/biostudies/arrayexpress/studies/E-MTAB-10065 | EMBL-EBI, E-MTAB-10065 |

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
