## [Editor Report · eLife Assessment]

This **important** study uses single-neuron Patch-seq RNA sequencing to investigate the process by which RNA editing can produce protein diversity and regulate function in various cellular contexts. The computational analyses of the data collected are **convincing**, and from an analytical standpoint, this paper is a notable advance in seeking to provide a biological context for massive amounts of data in the field. The study would be of interest to biologists looking at the effects of RNA editing in the diversification of cellular behaviour.

---

## [Referee Report · Reviewer #1 (Public review)]

The importance of RNA editing in producing protein diversity is a widespread process that can regulate how genes function in various cellular contexts. Despite the importance of the process, we still lack a thorough knowledge of the profile of RNA editing targets in known cells. Crane and colleagues take advantage of a recently acquired scRNAseq database for Drosophila type Ib and Is larval motoneurons and identify the RNA editing landscape that differs in those cells. They find both canonical (A  I) and non-canonical sites and characterize the targets, their frequencies, and determine some of the "rules" that influence RNA editing. They compare their database with existing databases to determine a reliance on the most well-known deaminase enzyme ADAR, determine the activity-dependence of editing profiles, and identify editing sites that are specific to larval Drosophila, differing from adults. The authors also identify non-canonical editing sites, especially in the newly appreciated and identified regulator of synaptic plasticity, Arc1.

The paper represents a strong analysis of recently made RNAseq databases from their lab and takes a notable approach to integrate this with other databases that have been recently produced from other sources. One of the places where this manuscript succeeds is in a thorough approach to analyzing the considerable amount of data that is out there regarding RNAseq in these differing motoneurons, but also in comparing larvae to adults. This is a strong advance. It also enables the authors to begin to determine rules for RNA editing. From an analytical standpoint, this paper is a notable advance in seeking to provide a biological context for massive amounts of data in the field. Further, it addresses some biological aspects in comparing WT and adar mutants to assess one potential deaminase, addresses activity-dependence, and begins to reveal profiles of canonical and non-canonical editing.

---

## [Referee Report · Reviewer #2 (Public review)]

Summary:

The study uses single-neuron Patch-seq RNA sequencing in two subgroups of Drosophila larval motoneurons (1s and 1b) and identifies 316 high-confidence canonical mRNA edit sites, which primarily (55%) occur in the coding regions of the mRNAs (CDS). Most of the canonical mRNA edits in the CDS regions include neuronal and synaptic proteins such as Complexin, Cac, Para, Shab, Sh, Slo, EndoA, Syx1A, Rim, RBP, Vap33, and Lap, which are involved in neuronal excitability and synaptic transmission. Of the 316 identified canonical edit sites, 60 lead to missense RNAs in a range of proteins (nAChRalpha5, nAChRalpha6, nAChRbeta1, ATPalpha, Cacophony, Para, Bsk, Beag, RNase Z) that are likely to have an impact on the larval motoneurons' development and function. Only 27 sites show editing levels higher than 90% and a similar editing profile is observed between the 1s and 1b motoneurons when looking at the number of edit sites and the fraction of reads edited per cell, with only 26 RNA editing sites showing a significant difference in the editing level. The variability of edited and unedited mRNAs suggests stochastic editing. The two subsets of motoneurons show many noncanonical editing sites, which, however, are not enriched for neuron-specific genes, therefore causing more silent changes compared to canonical editing sites. Comparison of the mRNA editing sites and editing rate of the single neuron Patch-seq RNA sequencing dataset to three other RNAseq datasets, one from same stage larval motoneurons and two from adult heads nuclei, show positive correlations in editing frequencies of CDS edits between the patch-sec larval 1b + 1s MNs and all other three datasets, with stronger correlations for previously annotated edits and weaker correlations for unannotated edits. Several of the identified editing targets are only present in the single neuron Patch-seq RNA sequencing dataset, suggesting cell-type-specific or developmental-specific editing. Editing appears to be resistant to changes in neuronal activity as only a few sites show evidence of being activity-regulated.

Strengths:

The study employs GAL4 driver lines available in the Drosophila model to identify two subtypes of motoneurons with distinct biophysical and morphological features. In combination with single-neuron Patch-seq RNA sequencing, it provides a unique opportunity to identify RNA editing sites and rates specific to specific motoneuron subtypes. The RNA seq data is robustly analysed, and high-confidence mRNA edit sites of both canonical and noncanonical RNA editing are identified.

The mRNA editing sites identified from the single neuron Patch-seq RNA sequencing data are compared to editing sites identified across other RNAseq datasets collected from animals at similar or different developmental stages, allowing for the identification of editing sites that are common to all or specific to a single dataset.

Weaknesses:

Although the analysed motoneurons come from two distinct subtypes, it is unclear from how many Drosophila larvae the motoneurons were collected and from which specific regions along the ventral nerve cord (VNC). Therefore, the study does not consider possible differences in editing rate between samples from different larvae that could be in different active states or neurons located at different regions of the VNC, which would receive inputs from slightly different neuronal networks.

The RNA samples include RNAs located both in the nucleus and the cytoplasm, introducing a potential compartmental mismatch between the RNA and the enzymes mediating the editing, which could influence editing rate. Similarly, the age of the RNAs undergoing editing is unknown, which may influence the measured editing rates.

---

## [Referee Report · Reviewer #3 (Public review)]

Summary:

The study consists of extensive computational analyses of their previously released Patch-seq data on single MN1-Ib and MNISN-Is neurons. The authors demonstrate the diversity of A>I editing events at single-cell resolution in two different neuronal cell types, identifying numerous A>I editing events that vary in their proportion, including those that cause missense mutations in conserved amino acids. They also consider "noncanonical" edits, such as C>T and G>A, and integrate publicly available data to support these analyses.

In general, the study contains a valuable resource to assess RNA editing in single neurons and opens several questions regarding the diversity and functional implications of RNA editing at single-cell resolution. The conclusions from the study are generally supported by their data; however, the study is currently based on computational predictions and would therefore benefit from experimentation to support their hypotheses and demonstrate the effects of the editing events identified on neuronal function and phenotype.

Strengths:

The study uses samples that are technically difficult to prepare to assess cell-type-specific RNA editing events in a natural model. The study also uses public data from different developmental stages that demonstrate the importance of considering cell type and developmental stage-specific RNA regulation. These critical factors, particularly that of developmental timing, are often overlooked in mechanistic studies.

Extensive computational analysis, using public pipelines, suitable filtering criteria, and accessible custom code, identifies a number of RNA editing events that have the potential to impact conserved amino acids and have subsequent effects on protein function. These observations are supported through the integration of several public data sets to investigate the occurrence of the edits in other data sets, with many identified across multiple data sets. This approach allowed the identification of a number of novel A>I edits, some of which appear to be specific to this study, suggesting cell/developmental specificity, whilst others are present in the public data sets but went unannotated.

The study also considers the role of Adar in the generation of A>I edits, as would be expected, by assessing the effect of Adar expression on editing rates using public data from adar mutant tissue to demonstrate that the edits conserved between experiments are mainly Adar-sensitive. This would be stronger if the authors also performed Patch-seq experiments in adar mutants to increase confidence in the identified edit sites.

Weaknesses:

Whilst the study makes interesting observations using advanced computational approaches, it does not demonstrate the functional implications of the observed editing events. The functional impact of the edits is inferred from either the nature of the change to the coding sequence and the amino acid conservation, or through integration of other data sets. Although these could indeed imply function, further experimentation would be required to confirm such as using their Alphafold models to predict any changes in structure. This limitation is acknowledged by the authors, but the overall strength of the interpretation of the analysis could be softened to represent this.

The study uses public data from more diverse cellular populations to confirm the role of Adar in introducing the A>I edits. Whilst this is convincing, the ideal comparison to support the mechanism behind the identified edits would be to perform patch-seq experiments on 1b or 1s neurons from adar mutants. However, although this should be considered when interpreting the data, these experiments would be a large amount of work and beyond the scope of the paper.

By focusing on the potential impact of editing events that cause missense mutations in the CDS, the study may overlook the importance of edits in noncoding regions, which may impact miRNA or RNA-binding protein target sites. Further, the statement that noncanonical edits and those that induce silent mutations are likely to be less impactful is very broad and should be reconsidered. This is particularly the case when suggesting that silent mutations may not impact the biology. Given the importance of codon usage in translational fidelity, it is possible that silent mutations induced by either A>I or noncanonical editing in the CDS impact translation efficiency. Indeed, this could have a greater impact on protein production and transcript levels than a single amino acid change alone.

---

## [Author Response]

**Reviewer #1:**

Indicated the paper provided a strong analysis of RNAseq databases to provide a biological context and resource for the massive amounts of data in the field on RNA editing. The reviewer noted that future studies will be important to define the functional consequences of the individual edits and why the RNA editing rules we identified exist. We address these comments below.

(1) The reviewer wondered about the role of noncanonical editing to neuronal protein expression.

Indeed, the role of noncanonical editing has been poorly studied compared to the more common A-to-I ADAR-dependent editing. Most non-canonical coding edits we found actually caused silent changes at the amino acid level, suggesting evolutionary selection against this mechanism as a pathway for generating protein diversity. As such, we suspect that most of these edits are not altering neuronal function in significant ways. Two potential exceptions to this were non-canonical edits that altered conserved residues in the synaptic proteins Arc1 and Frequenin 1. The C-to-T coding edit in the activity-regulated Arc1 mRNA that encodes a retroviral-like Gag protein involved in synaptic plasticity resulted in a P124L amino acid change (see Author response image 1 panel A below). ~50% of total Arc1 mRNA was edited at this site in both Ib and Is neurons, suggesting a potentially important role if the P124L change alters Arc1 structure or function. Given Arc1 assembles into higher order viral-like capsids, this change could alter capsid formation or structure. Indeed, P124 lies in the hinge region separating the N- and C-terminal capsid assembly regions (panel B) and we hypothesize this change will alter the ability of Arc1 capsids to assemble properly. We plan to experimentally test this by rescuing Arc1 null mutants with edited versus unedited transgenes to see how the previously reported synaptic phenotypes are modified. We also plan to examine the ability of the change to alter Arc1 capsid assembly in a collaboration using CyroEM.

**Author response image 1. sa4fig1:** A. AlphaFold predictions of Drosophila Arc1 and Frq1 with edit site noted. B. Structure of the Drosophila Arc1 capsid. Monomeric Arc1 conformation within the capsid is shown on the right with the location of the edit site indicated.

The other non-canonical edit (G-to-A) that stood out was in Frequenin 1 (Frq1), a multi-EF hand containing Ca^2+^ binding protein that regulates synaptic transmission, that resulted in a G2E amino acid substitution (location within Frq1shown in panel A above). This glycine residue is conserved in all Frq homologs and is the site of N-myristoylation, a co-translational lipid modification to the glycine after removal of the initiator methionine by an aminopeptidase. Myristoylation tethers Frq proteins to the plasma membrane, with a Ca^2+^-myristoyl switch allowing some family members to cycle on and off membranes when the lipid domain is sequestered in the absence of Ca^2+^. Although the G2E edit is found at lower levels (20% in Ib MNs and 18% in Is MNs), it could create a pool of soluble Frq1 that alters it’s signaling. We plan to functionally assay the significance of this non-canonical edit as well. Compared to edits that alter amino acid sequence, determining how non canonical editing of UTRs might regulate mRNA dynamics is a harder question at this stage and will require more experimental follow-up.

(2) The reviewer noted the last section of the results might be better split into multiple parts as it reads as a long combination of two thoughts.

We agree with the reviewer that the last section is important, but it was disconnected a bit from the main story and was difficult for us to know exactly where to put it. All the data to that point in the paper was collected from our own PatchSeq analysis from individual larval motoneurons. We wanted to compare these results to other large RNAseq datasets obtained from pooled neuronal populations and felt it was best to include this at the end of the results section, as it no longer related to the rules of RNA editing within single neurons. We used these datasets to confirm many of our edits, as well as find evidence for some developmental and neuron-specific cell type edits. We also took advantage of RNAseq from neuronal datasets with altered activity to explore how activity might alter the editing machinery. We felt it better to include that data in this final section given it was not collected from our original PatchSeq approach.

**Reviewer #2**:

Noted the study provided a unique opportunity to identify RNA editing sites and rates specific to individual motoneuron subtypes, highlighting the RNAseq data was robustly analyzed and high-confidence hits were identified and compared to other RNAseq datasets. The reviewer provided some suggestions for future experiments and requested a few clarifications.

(1) The reviewer asked about Figure 1F and the average editing rate per site described later in the paper.

Indeed, Figure 1F shows the average editing rate for each individual gene for all the Ib and Is cells, so we primarily use that to highlight the variability we find in overall editing rate from around 20% for some sites to 100% for others. The actual editing rate for each site for individual neurons is shown in Figure 4D that plots the rate for every edit site and the overall sum rate for that neuron in particular.

(2) The reviewer also noted that it was unclear where in the VNC the individual motoneurons were located and how that might affect editing.

The precise segment of the larvae for every individual neuron that was sampled by Patch-seq was recorded and that data is accessible in the original Jetti et al 2023 paper if the reader wants to explore any potential anterior to posterior differences in RNA editing. Due to the technical difficulty of the Patch-seq approach, we pooled all the Ib and Is neurons from each segment together to get more statistical power to identify edit sites. We don’t believe segmental identify would be a major regulator of RNA editing, but cannot rule it out.

(3) The reviewer also wondered if including RNAs located both in the nucleus and cytoplasm would influence editing rate.

Given our Patch-seq approach requires us to extract both the cytoplasm and nucleus, we would be sampling both nuclear and cytoplasmic mRNAs. However, as shown in Figure 8 – figure supplement 3 D-F, the vast majority of our edits are found in both polyA mRNA samples and nascent nuclear mRNA samples from other datasets, indicating the editing is occurring co-transcriptionally and within the nucleus. As such, we don't think the inclusion of cytoplasmic mRNA is altering our measured editing rates for most sites. This may not be true for all non-canonical edits, as we did see some differences there, indicating some non-canonical editing may be happening in the cytoplasm as well.

**Reviewer #3**:

indicated the work provided a valuable resource to access RNA editing in single neurons. The reviewer suggested the value of future experiments to demonstrate the effects of editing events on neuronal function. This will be a major effort for us going forwards, as we indeed have already begun to test the role of editing in mRNAs encoding several presynaptic proteins that regulate synaptic transmission. The reviewer also had several other comments as discussed below.

(1) The reviewer noted that silent mutations could alter codon usage that would result in translational stalling and altered protein production.

This is an excellent point, as silent mutations in the coding region could have a more significant impact if they generate non-preferred rare codons. This is not something we have analyzed, but it certainly is worth considering in future experiments. Our initial efforts are on testing the edits that cause predictive changes in presynaptic proteins based on the amino acid change and their locale in important functional domains, but it is worth considering the silent edits as well as we think about the larger picture of how RNA editing is likely to impact not only protein function but also protein levels.

(2) The reviewer noted future studies could be done using tools like Alphafold to test if the amino acid changes are predicted to alter the structure of proteins with coding edits.

This is an interesting approach, though we don’t have much expertise in protein modeling at that level. We could consider adding this to future studies in collaboration with other modeling labs.

(3) The reviewer wondered if the negative correlation between edits and transcript abundance could indicate edits might be destabilizing the transcripts.

This is an interesting idea, but would need to be experimentally tested. For the few edits we have generated already to begin functionally testing, including our published work with editing in the C-terminus of Complexin, we haven’t seen a change in mRNA levels causes by these edits. However, it would not be surprising to see some edits reducing transcript levels. A set of 5’UTR edits we have generated in Syx1A seem to be reducing protein production and may be acting in such a manner.

(4) The reviewer wondered if the proportion of edits we report in many of the figures is normalized to the length of the transcript, as longer transcripts might have more edits by chance.

The figures referenced by the reviewer (1, 2 and 7) show the number of high-confidence editing sites that fall into the 5’ UTR, 3’ UTR, or CDS categories. Our intention here was to highlight that the majority of the high confidence edits that made it through the stringent filtering process were in the coding region. This would still be true if we normalized to the length of the given gene region. However, it would be interesting to know if these proportions match the expected proportions of edits in these gene regions given a random editing rate per gene region length across the Drosophila genome, although we did not do this analysis.

(5) The reviewer noted that future studies could expand on the work to examine miRNA or other known RBP binding sites that might be altered by the edits.

This is another avenue we could pursue in the future. We did do this analysis for a few of the important genes encoding presynaptic proteins (these are the most interesting to us given the lab’s interest in the synaptic vesicle fusion machinery), but did not find anything obvious for this smaller subset of targets.

(6) The reviewer suggested sequence context for Adar could also be investigated for the hits we identified.

We haven’t pursued this avenue yet, but it would be of interest to do in the future. In a similar vein, it would be informative to identify intron-exon base pairing that could generate the dsDNA template on which ADAR acts.

(7) The reviewer noted the disconnect between Adar mRNA levels and overall editing levels reported in Figure 4A/B.

Indeed, the lack of correlation between overall editing levels and Adar mRNA abundance has been noted previously in many studies. For the type of single cell Patch-seq approach we took to generate our RNAseq libraries, the absolute amount of less abundant transcripts obtained from a single neuron can be very noisy. As such, the few neurons with no detectable Adar mRNA are likely to represent that single neuron noise in the sampling. Per the reviewer’s question, these figure panels only show A-to-I edits, so they are specific to ADAR.

(8) The reviewer notes the scale in Figure 5D can make it hard to visualize the actual impact of the changes.

The intention of Figure 5D was to address the question of whether sites with high Ib/Is editing differences were simply due to higher Ib or Is mRNA expression levels. If this was the case, then we would expect to see highly edited sites have large Ib/Is TPM differences. Instead, as the figure shows, the vast majority of highly-edited sites were in mRNAs that were NOT significantly different between Ib and Is (red dots in graph) and are therefore clustered together near “0 Difference in TPMs”. TPMs and editing levels for all edit sites can be found in Table 1, and a visualization of these data for selected sites is shown in Figure 5E.